# Plants interfere with non-self recognition of a phytopathogenic fungus via proline accumulation to facilitate mycovirus transmission

Du Hai[1,2,3], Jincang Li[1,2,3], Daohong Jiang [1,2,3], Jiasen Cheng [1,2], Yanping Fu[1,2], Xueqiong Xiao[1,2], Huanran Yin[4], Yang Lin[1,2], Tao Chen[1,2,3], Bo Li [1,2,3], Xiao Yu[1,2,3], Qing Cai[1], Wei Chen[4], Ioly Kotta-Loizou [5,6] & Jiatao Xie [1,2,3] ✉

Non-self recognition is a fundamental aspect of life, serving as a crucial mechanism for mitigating proliferation of molecular parasites within fungal populations. However, studies investigating the potential interference of plants with fungal non-self recognition mechanisms are limited. Here, we demonstrate a pronounced increase in the efficiency of horizontal mycovirus transmission between vegetatively incompatible *Sclerotinia sclerotiorum* strains *in planta* as compared to in vitro. This increased efficiency is associated with elevated proline concentration in plants following *S. sclerotiorum* infection. This surge in proline levels attenuates the non-self recognition reaction among fungi by inhibition of cell death, thereby facilitating mycovirus transmission. Furthermore, our field experiments reveal that the combined deployment of hypovirulent *S. sclerotiorum* strains harboring hypovirulence-associated mycoviruses (HAVs) together with exogenous proline confers substantial protection to oilseed rape plants against virulent *S. sclerotiorum*. This unprecedented discovery illuminates a novel pathway by which plants can counteract *S. sclerotiorum* infection, leveraging the weakening of fungal non-self recognition and promotion of HAVs spread. These promising insights provide an avenue to explore for developing innovative biological control strategies aimed at mitigating fungal diseases in plants by enhancing the efficacy of horizontal HAV transmission.

The ability to discriminate between self and non-self is an ubiquitous phenomenon in all domains of life. In filamentous fungi, this process is typically termed vegetative (or heterokaryon) incompatibility (VIC)[1,2], wherein contact between genetically incompatible fungal strains triggers a non-self recognition reaction preventing any subsequent hyphal fusion via numerous pathways including programmed cell death (PCD)[1,2]. Studies exploring the molecular mechanisms underpinning self/non-self recognition in filamentous

[1]National Key Laboratory of Agricultural Microbiology, Huazhong Agricultural University, Wuhan, Hubei, China. [2]The Provincial Key Lab of Plant Pathology of Hubei Province, College of Plant Science and Technology, Huazhong Agricultural University, Wuhan, Hubei, China. [3]Hubei Hongshan Laboratory, Wuhan, Hubei, China. [4]National Key Laboratory of Crop Genetic Improvement and National Center of Plant Gene Research (Wuhan), Huazhong Agricultural University, Wuhan, China. [5]School of Life and Medical Sciences, University of Hertfordshire, Hatfield, UK. [6]Department of Life Sciences, Imperial College London, London, UK. ✉e-mail: jiataoxie@mail.hzau.edu.cn

fungi have primarily focused on ascomycetes such as *Neurospora crassa*, *Podospora anserina*, *Cryphonectria parasitica*, and *Botrytis cinerea*[3–12]. The genetic loci determining self/non-self recognition are traditionally termed *het* (heterokaryon) or *vic* (vegetative incompatibility). Within a fungal species, strains possessing identical alleles at these loci can engage in hyphal anastomosis and are categorized within the same vegetative compatibility group (VCG). Conversely, strains harboring different alleles are incompatible and belong to different VCGs[1,9,13]. Genes involved in the self/non-self recognition reaction encode heterotrimeric guanine nucleotide-binding proteins (G proteins), proteins involved in generation and scavenging of reactive oxygen species (ROS), and proteins containing HET domains, nucleotide oligomerization domain-like receptors, and cell wall remodeling checkpoint regulators[5,11,12,14,15]. Fundamentally, the self/non-self reaction functions akin to a rudimentary innate immune system, restricting the transmission of deleterious factors, including mycoviruses, between fungal strains, and thereby enhancing organismal fitness in natural environments[11].

Mycoviruses are obligate parasites of fungi. Unlike viruses infecting plant and animals, mycoviruses lack an extracellular phase in their replication cycle, with only rare exceptions[16]. Mycoviruses are transmitted horizontally between fungal strains and vertically from parent to offspring, through respectively hyphal fusion or sporulation. While the majority of mycoviruses do not induce discernible phenotypic alterations in their host fungi and their infections often remain latent, few mycoviruses can modulate fungal traits such as virulence[17–19]. Moreover, mycoviruses seemingly associated with cryptic infections may exert roles that are not readily apparent through conventional experimental approaches, such as the mycovirus-mediated thermal tolerance observed in fungi and plants[18]. Therefore, mycoviruses potentially play critical ecological roles in natural ecosystems. Of particular interest are mycoviruses that confer hypovirulence to phytopathogenic fungi, garnering considerable attention for their potential application as biological control agents against fungal diseases afflicting trees and crops. For instance, Cryphonectria hypovirus 1 (CHV1) has been successfully utilized in Europe to control the phytopathogenic fungus, *Cryphonectria parasitica*, responsible for chestnut blight[20]. However, success has been inconsistent in other geographical regions, particularly in North America. It has been postulated that the intricate structure of *C. parasitica* VCGs might be one of the factors contributing to the failure of CHV1-mediated control of chestnut blight[20,21]. Hypovirulence-associated mycoviruses (HAVs) are especially noteworthy due to their ability to spread within populations of phytopathogenic fungi via fungal anastomosis. Therefore, the rapid spread of HAVs in natural host populations is crucial for successfully controlling plant diseases. However, successful dissemination of HAVs within natural host populations is contingent upon uninterrupted horizontal transmission facilitated by hyphal fusion. This process can be hindered by factors such as VIC[9,22,23], potentially limiting the efficacy of mycovirus-mediated biological control strategies[20].

Despite the prevalence of VIC within fungal species and the intricate structure of VCGs, mycoviruses are ubiquitous in major fungal groups, with individual fungal isolates often harboring diverse mycoviruses[17,24–27]. Studies conducted with *C. parasitica* and *Rosellinia necatrix* have demonstrated that the rates of horizontal mycovirus transmission are higher *in planta* than in vitro[28,29]. Furthermore, highly similar mycoviruses are sometimes found to infect different fungal species or genera in nature[26,30]. These observations suggest that VIC can be weakened, allowing horizontal mycovirus transmission even between distinct fungal genera. However, the underlying mechanisms driving this phenomenon remain unexplored.

Proline is a pivotal multi-functional amino acid known for its diverse biological functions across plants, mammals, and fungi. Constituting approximately 5% of the total pool of free amino acids in plants under normal conditions, proline proportion can surge to as much as 80% of the amino acid pool following stress[31]. Interestingly, proline accumulation is a hallmark plant response during the early stages of infection by phytopathogenic fungi[32–34]. In mammals, proline plays a protective role against $H_2O_2$-induced cell death and apoptosis, but it has also been shown to promote apoptosis in certain human colon cancer cell lines[35]. In fungi, proline serves as a potent antioxidant, effectively scavenging ROS and inhibiting apoptosis-like PCD[36]. Despite the established association between PCD and fungal non-self recognition, the involvement of proline in inhibiting non-self recognition between VIC groups has not been studied previously.

*S. sclerotiorum*, a cosmopolitan phytopathogenic fungus, exhibit a broad host range, infecting over 700 plant species and causing significant economic losses annually[37]. *S. sclerotiorum* harbors a diverse array of mycoviruses, some of which recognized as HAVs with potential as biological control agents due to their substantial impact on virulence[25,38–45]. Given that *S. sclerotiorum* does not produce typical asexual spores, RNA mycoviruses in *S. sclerotiorum* primarily rely on hyphal anastomosis for horizontal transmission. However, the VCGs of *S. sclerotiorum* exhibit notable complexity. For instance, a study of 30 individual *S. sclerotiorum* strains isolated from a single rapeseed field illustrated that they are associated with 27 different VCGs[46]. Therefore, the complex *S. sclerotiorum* VCGs pose a potential impediment to horizontal mycovirus transmission and consequently challenge the efficacy of HAV-based biocontrol applications. Nevertheless, the fact that *S. sclerotiorum* strains isolated from their natural habitat harbor diverse mycoviruses indicates the existence of mechanisms facilitating mycovirus transmission in nature and suggests that even fungal populations with a high degree of VIC are amenable to mycovirus-based control strategies[25,27,38].

In this study, our primary objective was to investigate how mycoviruses overcome limitations in effective horizontal transmission imposed by the fungal non-self recognition system in natural environments. Our findings reveal that the transmission frequency of mycoviruses between *S. sclerotiorum* VIC strains is markedly higher *in planta* as compared to in vitro, concomitant with an increase in plant proline levels upon *S. sclerotiorum* infection. We present compelling evidence that plant proline serves to attenuate non-self recognition reactions in *S. sclerotiorum*, thus facilitating horizontal mycovirus transmission. By elucidating the mechanism underlying elevated horizontal mycovirus transmission levels *in planta*, our study offers valuable insights and proposes a straightforward yet innovative strategy for enhancing the application of HAVs in plant disease management in agricultural settings.

## Results

### Mycovirus transmission between *S. sclerotiorum* strains is enhanced in plants

*S. sclerotiorum* strain Ep-1PN, co-infected with Sclerotinia sclerotiorum debilitation-associated RNA virus (SsDRV) and Sclerotinia sclerotiorum RNA virus L (SsRV-L), exhibited several phenotypic traits associated with hypovirulence, including abnormal colony morphology, reduced growth rate, and decreased virulence on oilseed rape (*Brassica napus*) plants (Fig. 1A, B)[43,47]. Strain Ep-1PNA367 is a virulent HAVs-free strain, obtained from HAVs-infected Ep-1PN through a single ascospore isolation and therefore vegetatively compatible with Ep-1PN. Co-culturing of Ep-1PN and Ep-1PNA367 on Potato Dextrose Agar (PDA) resulted in the unrestricted horizontal transmission of mycoviruses from Ep-1PN (donor strain) to Ep-1PNA367 (recipient strain), converting Ep-1PNA367 into a hypovirulent strain as evidenced by its abnormal colony morphology and reduced growth rate (Fig. 1A)[43]. Conversely, mycoviruses were rarely transmitted on PDA from Ep-1PN to incompatible strains 1980m,

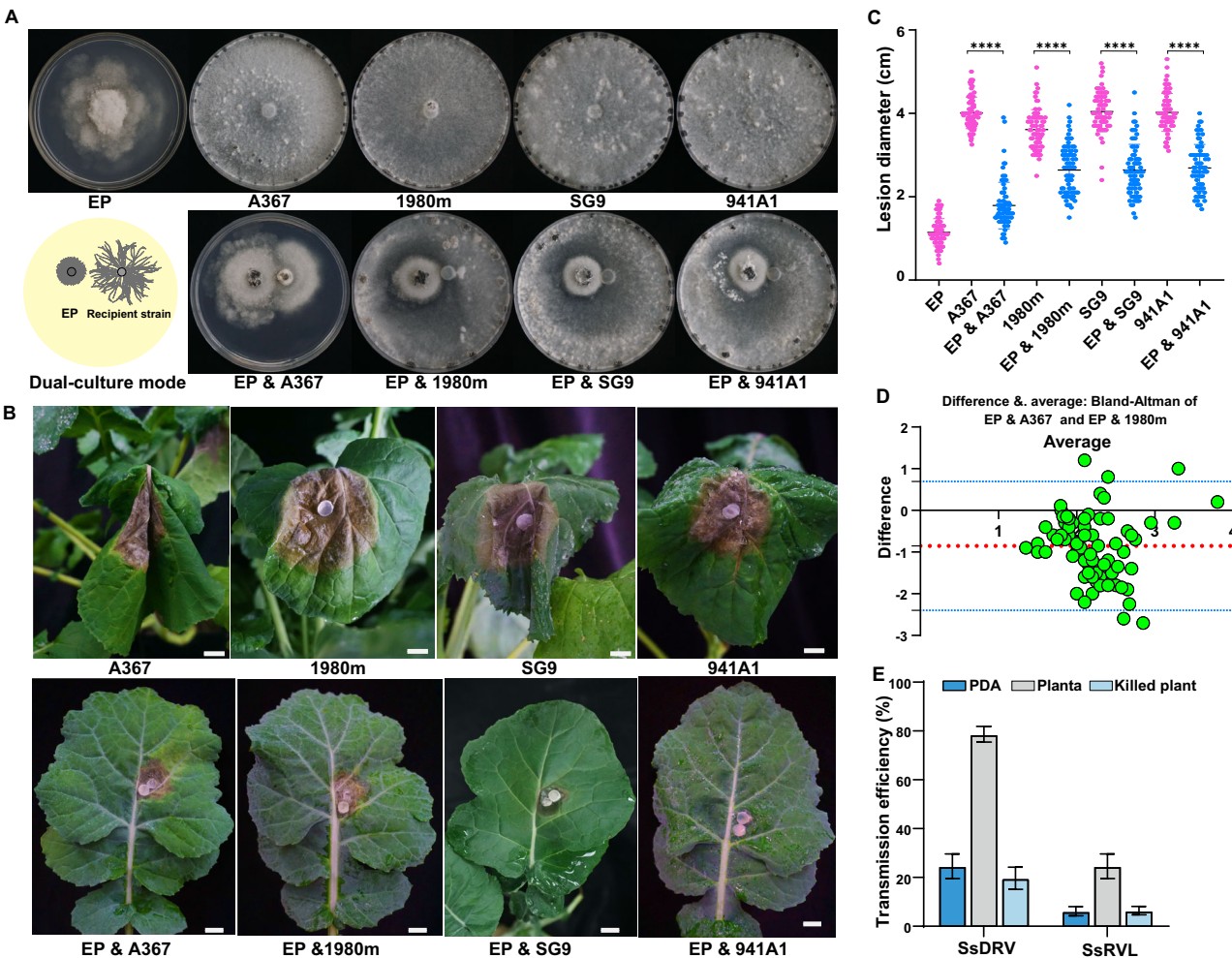

**Fig. 1 | Plant enhances mycovirus transmission between VIC *S. sclerotiorum* strains. A** Colony morphology of *S. sclerotiorum* strains Ep-1PN (EP), Ep-1PNA367 (A367), 1980m, SG9, and SCH941A1 (941A1) at 7 dpi (upper panel), and co-cultures of the HAVs-infected hypovirulent strain Ep-1PN as donor and the other four HAVs-free virulent strains as recipients for 5 dpi (lower panel) on PDA at 20 °C. Ep-1PN is compatible with its single-ascospore derivative Ep-1PNA367, and incompatible with the other three strains. **B** Virulence assay of Ep-1PNA367, 1980m, SG9, and SCH941A1 (upper panel), and their co-inoculations with Ep-1PN (lower panel) on living oilseed rape plants (72 hpi, 20 °C). The white bar represents 1 cm. **C** Diameters of lesions caused by Ep-1PN, Ep-1PNA367, 1980m, SG9, and SCH941A1 alone, and following co-inoculation of those strains with Ep-1PN on living oilseed rape plants (72 hpi, 20 °C), n = 76 biologically independent samples. **D** Analysis of diameters of

lesions caused by co-inoculation of Ep-1PN with either Ep-1PNA367 or 1980m using the Bland-Altman method, illustrating that only 6.8% (5/74) of lesion size values were outside the 95% consistency limit (95% limits of agreement) of the maximum error range. **E** Horizontal transmission efficiency of two mycoviruses, SsDRV and SsRVL, from Ep-1PN to 1980m on PDA, living oilseed rape plants, and oilseed rape leaves previously stored in −80 °C (n = 3 and N = 40; n = 3 indicates that we conducted three independent biological repeats and N = 40 indicates that we performed 40 co-cultures for each biological repeat, with each recipient strain from the 40 co-cultures sub-cultured on plates containing hygromycin or neomycin and used to measure mycovirus transmission efficiency). Data are presented as mean ± SD, two-tailed Student's t test. ****p < 0.0001. Source data are provided as a Source Data file.

SG9, and SCH941A1, as illustrated by the lack of traits associated with hypovirulence in the recipient strains (Fig. 1A). On oilseed rape plants, the HAVs-infected strain Ep-1PN caused significantly smaller lesions as compared to the four virulent strains. Notably, co-inoculation of all four virulent strains with Ep-1PN resulted in significantly smaller lesions as compared to the virulent strains alone (Fig. 1B–D). These observations suggested that, *in planta*, mycoviruses were horizontally transmitted from Ep-1PN to the virulent strains despite VIC, thereby conferring hypovirulence.

To test this hypothesis, we re-isolated incompatible strain 1980m, which is labeled with a hygromycin resistance marker, from plant lesions through three rounds of sub-culturing on PDA with hygromycin (Supplementary Fig. 1). Subsequent reverse transcription followed by polymerase chain reaction (RT-PCR) confirmed the presence of mycoviruses SsDRV and SsRVL in 1980m, with the transmission frequency being significantly higher in planta (79% for

SsDRV and 25% for SsRVL) than on PDA (25% for SsDRV and 6% for SsRVL) (Fig. 1E). This increased transmission rate was only observed when experiments were conducted on fresh leaves and not leaves previously stored in −80 °C (Fig. 1E and Supplementary Fig. 1), indicating that the leaves must be metabolically active to influence transmissibility. Therefore, we concluded that oilseed rape plants can enhance horizontal transmission of mycoviruses between incompatible strains.

## Fungal VIC-associated genes are downregulated *in planta*
To unravel the potential mechanism underlying enhanced mycovirus transmissibility *in planta*, strain 1980m was co-cultured separately with HAVs-infected strain Ep-1PN and HAVs-free strain Ep-1PNA367 on PDA and plants. Subsequently, genes related to the self/non-self recognition reaction and the G protein signaling pathway were identified in the *S. sclerotiorum* genome and their in vitro and

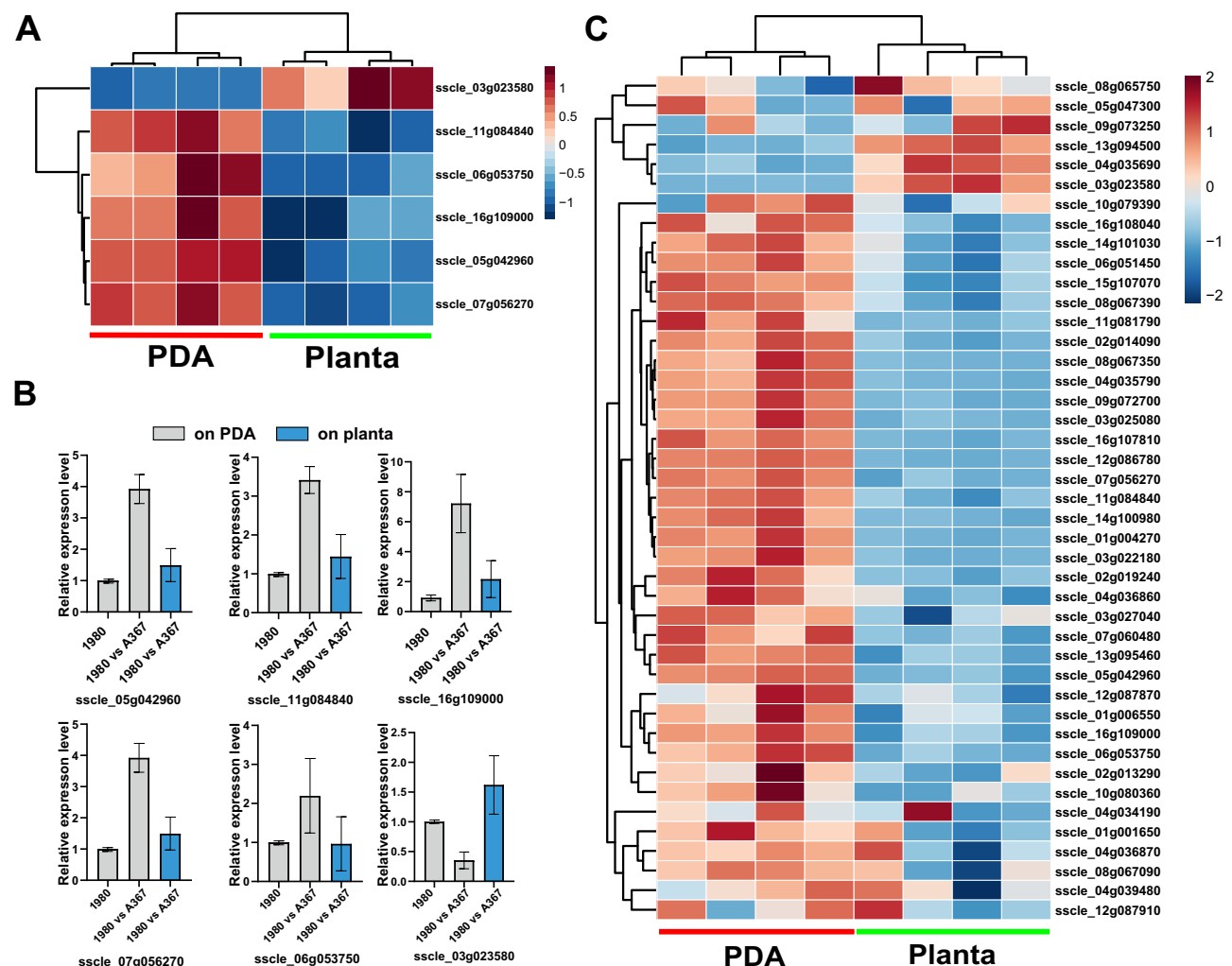

**Fig. 2 | Plant suppresses expression of fungal G protein and *vic*-related genes upon *S. sclerotiorum* infection. A** Expression cluster analysis of *S. sclerotiorum* genes encoding G protein subunits in 1980 and Ep-1PN co-cultures in vitro and *in planta*. **B** Expression levels of genes encoding G protein subunits in 1980 alone, and 1980 and Ep-1PN co-culture in vitro and *in planta*. Data are presented as mean ± SD, two-tailed Student's *t* test. *n* = 3 biologically repetitions. **C** Expression cluster analysis of *S. sclerotiorum* genes encoding proteins containing HET conserved domains in 1980 and Ep-1PN co-cultures in vitro and *in planta*. The relative expression of selected genes was plotted based on the threshold of RPKM value in RNA-seq data (Bioproject Accession: PRJNA908905). Red and blue indicate respectively high and low expression levels. Source data are provided as a Source Data file.

*in planta* expression levels were compared (Fig. 2 and Supplementary Fig. 2). In total, forty genes encoding proteins with a conserved HET domain were identified, the majority of which were downregulated *in planta* as compared to in vitro (Fig. 2C and Supplementary Fig. 2). Additionally, six genes encoding G proteins, including three *Gα* (sscle_11g084840, sscle_05g042960, and sscle_16g109000), two *Gβ* (sscle_07g056270 and sscle_03g023580), and one *Gγ* (sscle_06g053750), were identified. All except *Gβ2* were downregulated *in planta* as compared to in vitro (Fig. 2A, B and Supplementary Fig. 2A). Notably, the presence of HAVs in EP-1PN did not appear to have a significant effect on the expression levels of the aforementioned genes (Supplementary Fig. 2). Therefore, the attenuation of the VIC reaction is attributable to plant-fungus interactions, rather than mycovirus infection. We further compared the expression levels of genes encoding G proteins in other phytopathogenic fungi during growth in vitro and infection *in planta* by utilizing publicly available transcriptome data (Supplementary Fig. 3). Our analysis revealed that the majority of these genes were downregulated *in planta* as compared to their in vitro expression, suggesting that their suppression during plant-fungus interactions may be universal.

## Plant proline accumulation induced by fungal infection enhances mycovirus transmission

Proline levels are partly influenced by pyrroline-5-carboxylate (P5C) levels, its precursor molecule synthesized enzymatically by pyrroline-5-carboxylate synthase (P5CS)[48,49]. Proline levels increased respectively 4-fold and 10-fold in response to infection of oilseed rape with HAVs-infected hypovirulent strain Ep-1PN and HAVs-free virulent strain Ep-1PNA367 (Fig. 3A). It appears that this increase in proline levels is a response to fungal infection and is therefore more pronounced in the presence of the virulent strain that induces more severe infection as compared to the isogenic hypovirulent strain. A similar proline increase was also observed in *Arabidopsis thaliana* Col-0 plants during *S. sclerotiorum* infection (Fig. 3B). Plants normally possess two *p5cs* genes, and *p5cs2* but not *p5cs1* expression levels were upregulated in both oilseed rape and *A. thaliana* Col-0 plants infected with Ep-1PNA367 (Supplementary Fig. 4A), indicating that increased proline

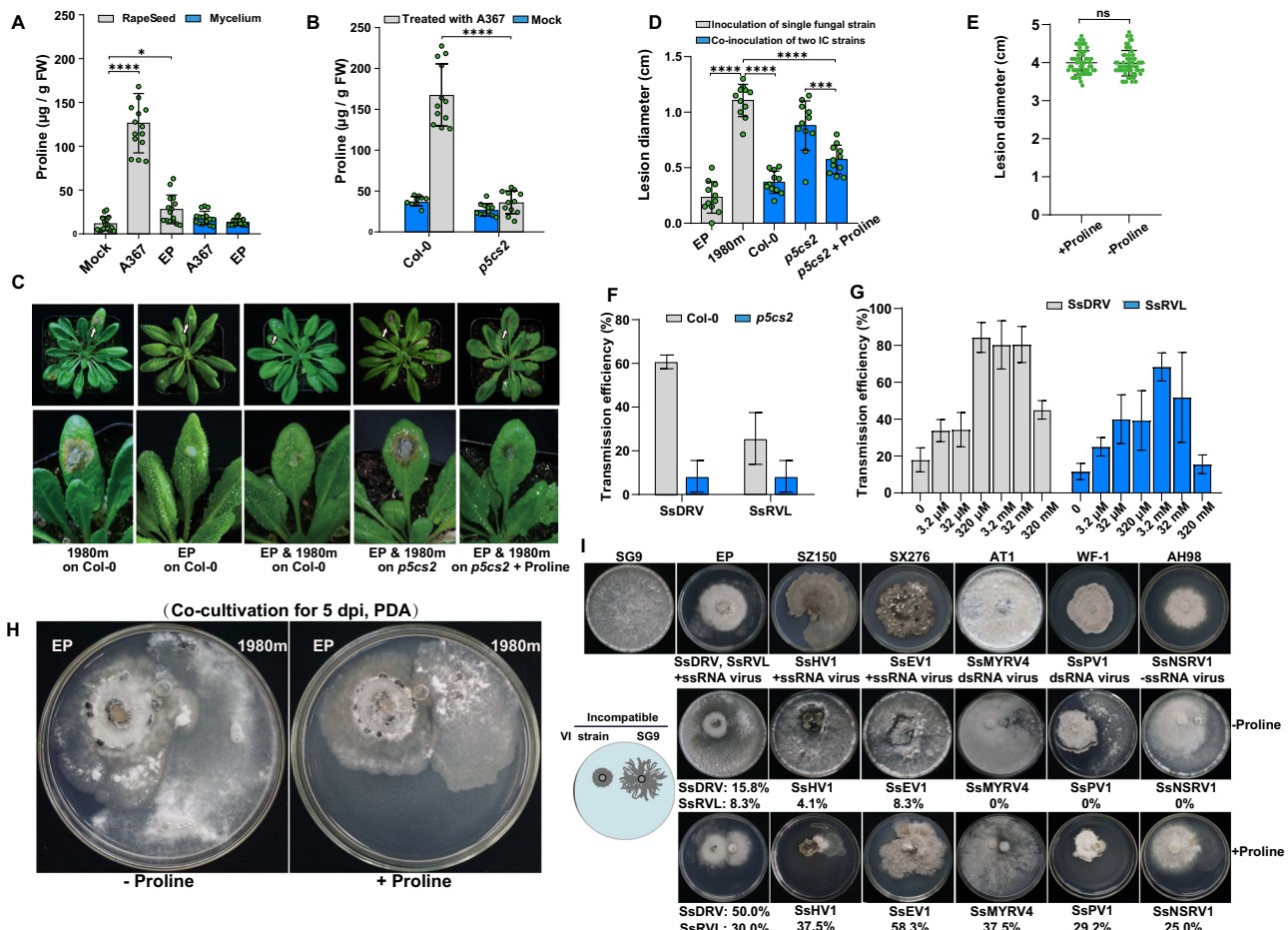

**Fig. 3 | Plant-derived proline increases upon *S. sclerotiorum* infection and promotes mycovirus transmission. A** Proline content of oilseed rape plants inoculated by HAVs-free virulent strain Ep-1PNA367 (A367) and HAVs-infected hypovirulent strain Ep-1PN (EP) at 24 hpi, and of Ep-1PNA367 and Ep-1PN mycelia (*n* = 15). **B** Proline content of *A. thaliana* Col-0 wildtype and *p5cs2* mutant inoculated by Ep-1PNA367 at 24 hpi (*n* = 11). **C** Virulence assay of strains 1980m, Ep-1PN, and Ep-1PN and 1980m co-inoculation (Ep & 1980m) on *A. thaliana* Col-0 wildtype and *p5cs2* mutant, and virulence assay of Ep & 1980m co-inoculation on the *p5cs2* mutant following spraying with exogenous proline (36 hpi, 20 °C) (*n* = 14). The lesions indicated by white arrows in the upper panel was enlarged in the lower panel. **D** Diameters of lesions caused by Ep-1PN and 1980m co-culture on *A. thaliana* Col-0 wildtype, *p5cs2* mutant, and *p5cs2* mutant plus proline (36 hpi, 20 °C) (*n* = 11). **E** Diameters of lesions caused by 1980m with or without proline on oilseed rape (72 hpi, 20 °C), illustrating that proline addition does not induce plant resistance to *S. sclerotiorum* infection (*n* = 57). **F** Transmission efficiency of two mycoviruses, SsDRV and SsRVL, from Ep-1PN to 1980m on *A. thaliana* Col-0 wildtype and *p5cs2* mutant (*n* = 3 for independent experiments, *N* = 8 for co-cultures). **G** Transmission efficiency of two mycoviruses, SsDRV and SsRVL, from Ep-1PN to

1980m on PDA containing proline at concentrations ranging from 3.2 μM to 320 mM (*n* = 3 for independent experiments, *N* = 12 for co-cultures). **H** Co-culture of Ep-1PN with 1980m on PDA (left) and PDA + P (right) (*n* = 16). **I** Upper panel, colony morphology of virus-free virulent SG9 (recipient) and six virus-infected hypovirulent strains SZ-150, SX276, Ep-1PNA367T1, WF-1, and AH98 (donors) on PDA or PDA + P for 7 days at 20 °C and corresponding mycovirus transmission efficiency (*n* = 24). The hypovirulent strains were infected by diverse mycoviruses, including +ssRNA mycoviruses (SsDRV, SsRVL, SsEV1, and SsHV1), dsRNA mycoviruses (SsPV1 and SsMYRV4), and −ssRNA mycoviruses (SsNSRV1), as indicated below each fungal colony. Lower panel, co-culture model diagram of SG9 and mycovirus-infected (VI) hypovirulent strains on PDA (left). Strain SG9 was co-cultured with six hypovirulent strains on PDA or PDA + P, and the associated mycovirus transmission efficiency is shown below for each fungal strain (*n* = 24). The transmission efficiency of all mycoviruses was increased on PDA + P. Data are presented as mean values ± SD, two-tailed Student's *t* test, *n* represents biologically independent samples, \**p* < 0.05; \*\*\**p* < 0.001; \*\*\*\**p* < 0.0001; ns not significant. Source data are provided as a Source Data file.

stems at least partially from upregulation of the enzyme responsible for synthesizing its precursor molecule.

Non-self recognition during hyphal fusion often coincides with accumulation of reactive oxygen species (ROS) and initiation of programmed cell death (PCD)[1,5]. Proline is known to act as a ROS scavenger and inhibit apoptosis-like PCD[36,50–52]. Therefore, we hypothesized that the enhanced mycovirus transmission between incompatible fungal strains observed *in planta* is facilitated by proline produced by the plant. To test this hypothesis, an *A. thaliana* Col-0 mutant with a T-DNA insertion in *p5cs2* leading to impaired proline production during *S. sclerotiorum* infection (Fig. 3B) was used in phenotypic comparisons with wildtype. Co-inoculation of

wildtype and *p5cs2* mutant plants with Ep-1PN and 1980m revealed significantly larger lesions in the *p5cs2* mutant as compared to wildtype, while exogenous proline application mitigated the increased susceptibility of *p5cs2* mutant plants (Fig. 3C, D). Subsequently, 1980m was re-isolated from lesions of wildtype and *p5cs2* mutant plants and HAVs infection was confirmed with the transmission frequency being significantly lower in the *p5cs2* mutant as compared to wildtype (Fig. 3F). Importantly, reduction in disease severity was observed only in the presence of both proline and the HAVs-infected strain (Fig. 3E), supporting the notion that proline enhances mycovirus transmission rather than inducing plant resistance against the pathogenic fungus.

## Proline provided exogenously enhances mycovirus transmission

To provide further evidence that proline enhances mycovirus transmission and to explore potential mechanisms in a tractable in vitro system, we established fungal cultures on PDA supplemented with proline in concentrations ranging from 3.2 μM to 320 mM, where Ep-1PN and 1980m exhibited robust growth. In this in vitro system, the highest transmission frequency of SsDRV and SsRVL from Ep-1PN to 1980m was observed at 3.2 mM proline (Fig. 3G), a concentration subsequently used for all further experiments and designated as PDA + P. Moreover, increased hyphal branching of 1980m was noted at its margin within the contact area (Fig. 3H). Together, these results suggest that proline, whether produced by plants or applied exogenously, can directly facilitate mycovirus transmission between incompatible *S. sclerotiorum* strains.

To evaluate whether this phenomenon is a widespread occurrence in *S. sclerotiorum*, we measured the horizontal transmission rates of HAVs in six previously reported hypovirulent strains, namely SZ-150, SX276, Ep-1PN, WF-1, AH98, and SCH733 (Supplementary Table 1 and Fig. 3I), to the incompatible strains SG9 and 1980m. Each HAV-infected donor strain was co-cultured with the each virus-free recipient strain on PDA and PDA + P. Across all combinations, the addition of proline resulted in increased HAVs transmission from donors to recipients, irrespective of whether the mycoviruses comprised dsRNA or ssRNA genomes (Fig. 3I and Supplementary Fig. 5). This finding suggests that the proline-mediated facilitation of mycovirus transmission is a broad-spectrum phenomenon in *S. sclerotiorum*.

## Proline promotes hyphal fusion, inhibits expression of VIC-associated genes, and reduces ROS accumulation in fungi

To investigate the cellular, biochemical and molecular mechanisms underpinning the proline-mediated enhancement of horizontal mycovirus transmission, we first examined the impact of proline on hyphal fusion between incompatible strains, particularly the induction of PCD associated with VIC reactions[1,9,53]. Following Evans blue assay and scanning electron microscopy (SEM) visualization, a typical necrotic zone was clearly observed at the interface between 1980m and incompatible strains Ep-1PNA367 or SCH733 on PDA but was significantly attenuated on PDA + P (Fig. 4A and Supplementary Figs. 5A and 6). Importantly, this necrotic zone was not at all evident between compatible strains Ep-1PN and Ep-1PNA367 (Fig. 1A). This observation, together with the fact that the mycovirus transmission efficiency should theoretically be 100% between two compatible strains[41] and therefore higher than that of two incompatible strains on PDA + P (Fig. 3G), suggest that proline weakens but do not completely eliminates the PCD response between incompatible strains. Examination of the interface between 1980m and its incompatible strains by transmission electron microscopy (TEM) revealed organelle degradation and extensive cytoplasmic vacuole formation on PDA, whereas cytoplasmic vacuole formation was reduced, and hyphal fusion was enhanced on PDA + P (Fig. 4B). Propidium iodide (PI) staining, used for detecting dead cells, further confirmed cell death during co-culture of 1980G and Ep-1PNA367G on PDA, whereas no visible PI staining was evident on PDA + P, indicating the absence of cell death (Fig. 4C). Confocal fluorescence microscopy allowed visualization of hyphal fusion between 1980m and Ep-1PNA367G, evident on PDA + P but not on PDA (Fig. 4D).

Subsequently, we conducted transcriptome profiling of two incompatible *S. sclerotiorum* strains on PDA, PDA + P and *in planta*. In total, 1285 differentially expressed genes (DEGs; >2-fold level), of which 1595 and 690 were respectively downregulated and upregulated, were noted on PDA + P as compared to PDA (Fig. 4E). Gene Ontology (GO) "biological process" and "cellular component" terms found to be enriched in the upregulated genes included cell wall organization or biogenesis, vacuole, cell wall, membrane, and extracellular region (Supplementary Fig. 8). Conversely, GO "biological process" terms found to be enriched in the downregulated genes included transport (transmembrane, lipid, ion, etc.), response to environmental factors (oxidative stress, heat, chemical, and DNA damage stimulus, etc.), and cell wall remodeling (Fig. 4F). Consistent with observations *in planta* (Fig. 2), expression levels of genes encoding HET and G proteins were predominantly downregulated on PDA + P as compared to PDA (Fig. 4G). However, no significant change in the expression levels of these genes was noted between PDA and PDA + P when co-culturing two compatible *S. sclerotiorum* strains (Supplementary Fig. 7), suggesting that no PCD reaction occurred in this case as expected based on our observations at the microscopic level. The expression levels of two ROS-related genes were induced upon contact of incompatible strains on PDA, but this induction was inhibited on PDA + P (Fig. 4H). Additionally, nitroblue tetrazolium (NBT) staining showed that proline inhibits ROS accumulation in 1980m exposed to Ep-1PNA367 supernatant following lysis (Fig. 4I).

## Fungal VIC-associated genes downregulated by proline impede mycovirus transmission

To examine whether any of the genes whose expression is downregulated by proline in vitro and/or *in planta* play a role in horizontal mycovirus transmission, we generated deletion mutants for a subset of them in the genetic background of strain 1980. This deletion mutant analysis encompassed genes *Ssvic1* to *Ssvic6*, *Ssvib1*, and *Sscwr1* to *Sscwr3*, which encode proteins containing HET/NT80 or LMPO conserved domains (Supplementary Figs. 9 and 10). *Ssvic2* deletion resulted in reduced growth, while *Ssvib1* deletion affected sclerotia development, including their morphology and numbers. The remaining deletions exhibited colony morphology and growth rates similar to their parent strain (Fig. 4J). Subsequently, Ep-1PN (donor) was co-cultured with 1980m or individual deletion mutants (recipients) on PDA (Fig. 4J). The hyphae at the margin of most deletion mutants serving as recipient strains displayed excessive branching and reduced growth, particularly in the case of Δ*Ssvic3*. The recipient strains were then subcultured on neomycin-containing PDA for three generations to fully eliminate the donor and the transmission rate of mycoviruses was assessed using RT-PCR. Transmission rates were significantly increased in Δ*Ssvic1*, Δ*Ssvic3*, Δ*Ssvic4*, Δ*Ssvic6*, Δ*Sscwr1*, and Δ*Sscwr2* as compared to their parent strain (Fig. 4J). Remarkably, approximately 75% of the Δ*Ssvic3* sub-isolates tested exhibited phenotypic traits associated with hypovirulence, consistent with the SsDRV transmission rate (Fig. 4J).

## Exogenous proline enhances the biocontrol efficiency of HAVs and increases oilseed rape yield

To ascertain the applicability of laboratory findings for enhanced control of Sclerotinia disease using HAVs in conjugation with exogenous proline, we conducted field experiments on oilseed rape for a period of over 2 years, from 2021 to 2022. In 2021, the disease incidence decreased by 41.9% and yield increased by 11.2% following combined Ep-1PN and proline treatment, while disease incidence decreased by 26.9% and yield increased by 4.8% following treatment with Ep-1PN alone as compared to non-treated controls (Fig. 5). Similar results were observed in the field experiments conducted in 2022.

## Discussion

Fungal non-self recognition mechanisms are commonly regarded as the primary barrier to horizontal mycovirus transmission. However, our understanding of how mycoviruses are horizontally transmitted between fungi largely comes from in vitro experiments, with limited insights into whether plants possess mechanisms enhancing mycovirus transmission between pathogenic fungi, thus decreasing

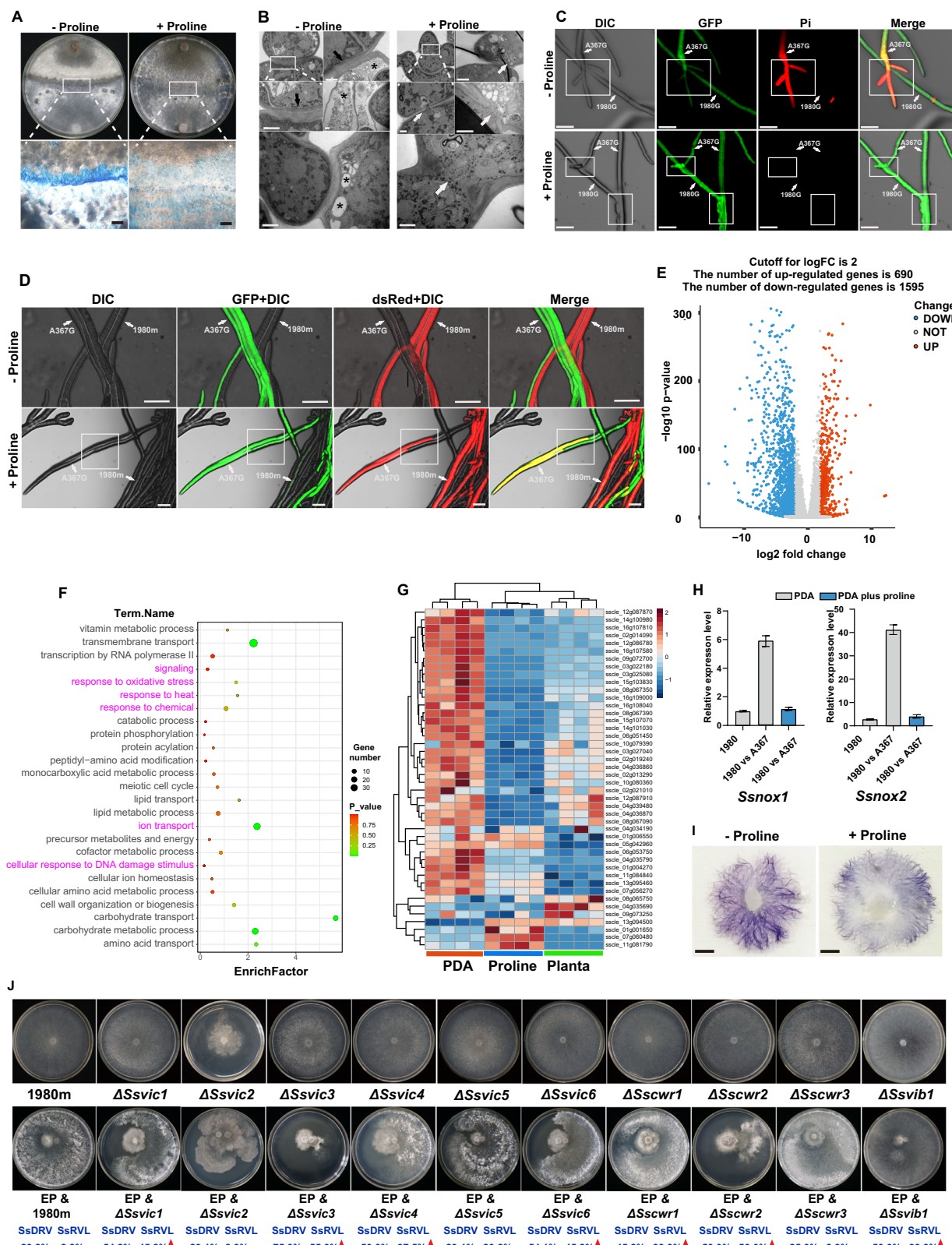

disease severity. Moreover, in vitro experiments do not fully capture the natural process of horizontal mycovirus transmission in plant-associated fungi, as they overlook the important factor of "plant". Therefore, we investigated the impact of host plants (oilseed rape and *Arabidopsis*) on mycovirus transmission between incompatible strains of a phytopathogenic fungus (*S. sclerotiorum*). Our findings revealed a significant increase in the transmission rate of

mycoviruses *in planta* as compared to in vitro. This phenomenon mirrors observations in other mycovirus-fungus-plant interaction systems. For example, the transmission rate of the prototypic CHV1 between incompatible strains of *C. parasitica* on trees consistently surpasses that observed in in vitro experiments[28], and mycovirus transmission can occur between incompatible *R. necatrix* strains on apple branches[29]. Plants appear to attenuate VIC response in

**Fig. 4 | Proline alleviates the fungal non-self recognition reaction by inhibiting *vic*-related gene expression and ROS accumulation. A** Evans blue staining of two incompatible strains 1980m and SCH733 co-cultured on PDA and PDA + P, for the detection of the necrotic zone and PCD response at their interface. The interface region indicated by the white frame in the upper panel was enlarged in the lower panel. Scale bar, 3 mm. **B** TEM visualization of cell fusion between Ep-1PNA367G and 1980m on PDA and PDA + P. The fusion pore is indicated by white arrows. The vacuole formation and organelle degradation are indicated by asterisks, and cell wall thickening is indicated by black arrows. Scale bar, 1 μm. **C** PI staining and visualization using confocal microscopy for detection of the cell death at the interface between 1980m and Ep-1PNA367G on PDA and PDA + P. The white box indicates regions where apoptosis occurs in the two incompatible strains stained by PI, while the addition of proline significantly reduces the extent of apoptosis in these regions. Scale bar, 50 μm. **D** Ep-1PNA367G and 1980m interface on PDA and

PDA + P visualized under confocal microscopy; the orientation of the arrows indicates that the hyphae originally labeled with one fluorescent group showed a signal of another fluorescent group, suggesting the fusion of hyphae carrying different fluorescent labels. Scale bar, 50 μm. **E** Volcano plot showing the number of DEGs in co-cultured VIC strains on PDA and PDA + P. **F** GO enrichment analysis of down-regulated DEGs from (**E**). The pathways marked by magenta are closely related to PCD or ROS. **G** Heatmap showing the expression levels of *vic*-related genes when two VIC strains interface on PDA, PDA + P, and oilseed rape plants. **H** Expression levels of genes related to G proteins and ROS in strain 1980 alone, and co-cultured with Ep-1PNA367 on PDA and PDA + P, as shown by RT-qPCR. Data are mean ± SD of *n* = 3 independent experiments. **I** NBT staining of 1980m treated with supernatant from Ep-1PNA367G. Scale bar, 0.5 cm. **J** Transmission efficiency of SsDRV and SsRVL from Ep-1PN to 1980m or deletion transformants (*n* = 24 for co-cultures). Source data are provided as a Source Data file.

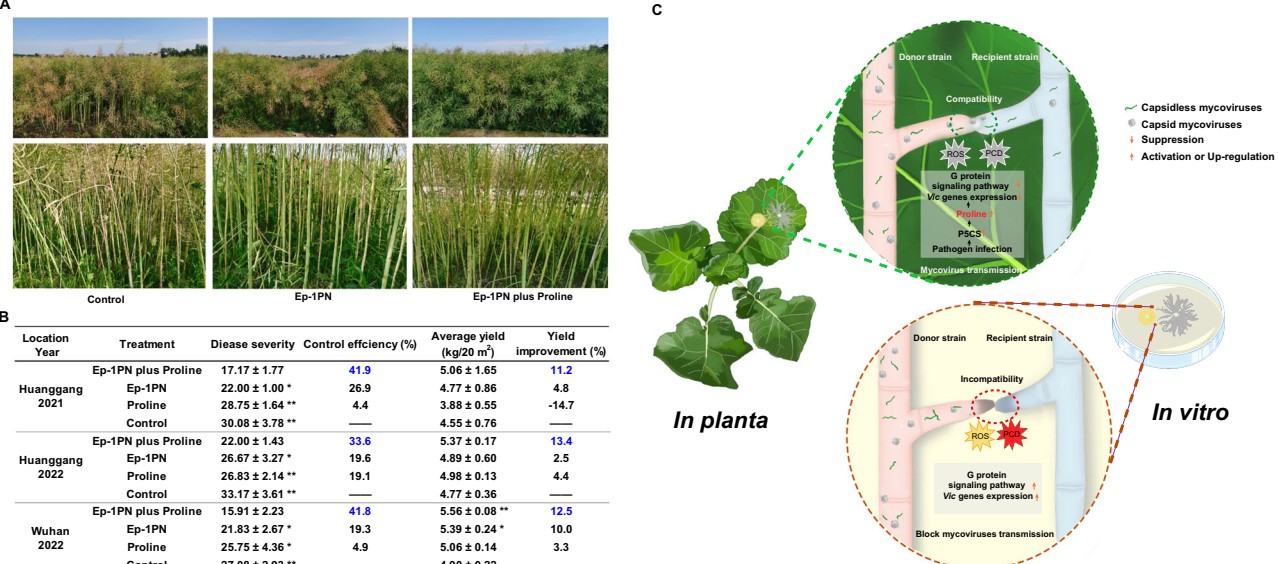

**Fig. 5 | Proline enhances the biological efficacy of HAVs against Sclerotinia stem rot of oilseed rape and improves yield. A** Symptoms of Sclerotinia stem rot on oilseed rape after spraying with hyphal fragment suspensions of Ep-1PN with and without proline. In the upper panel, grayish-yellow colors signify diseased plants; in the lower panel, grayish-yellow discoloration of stems is evident and caused by *S. sclerotiorum*. **B** Application of Ep-1PN with and without proline improved yield and suppressed stem rot in field experiments as compared to controls (proline and

| Location Year | Treatment | Diease severity | Control effciency (%) | Average yield (kg/20 m²) | Yield improvement (%) |
|---|---|---|---|---|---|
| Huanggang 2021 | Ep-1PN plus Proline | 17.17 ± 1.77 | 41.9 | 5.06 ± 1.65 | 11.2 |
| | Ep-1PN | 22.00 ± 1.00 * | 26.9 | 4.77 ± 0.86 | 4.8 |
| | Proline | 28.75 ± 1.64 ** | 4.4 | 3.88 ± 0.55 | -14.7 |
| | Control | 30.08 ± 3.78 ** | —— | 4.55 ± 0.76 | —— |
| Huanggang 2022 | Ep-1PN plus Proline | 22.00 ± 1.43 | 33.6 | 5.37 ± 0.17 | 13.4 |
| | Ep-1PN | 26.67 ± 3.27 * | 19.6 | 4.89 ± 0.60 | 2.5 |
| | Proline | 26.83 ± 2.14 ** | 19.1 | 4.98 ± 0.13 | 4.4 |
| | Control | 33.17 ± 3.61 ** | —— | 4.77 ± 0.36 | —— |
| Wuhan 2022 | Ep-1PN plus Proline | 15.91 ± 2.23 | 41.8 | 5.56 ± 0.08 ** | 12.5 |
| | Ep-1PN | 21.83 ± 2.67 * | 19.3 | 5.39 ± 0.24 * | 10.0 |
| | Proline | 25.75 ± 4.36 * | 4.9 | 5.06 ± 0.14 | 3.3 |
| | Control | 27.08 ± 2.93 ** | —— | 4.90 ± 0.22 | —— |

water). **C** The putative model for plant-derived proline-mediated inhibition of fungal non-self recognition reaction to promote mycovirus transmission. The interaction between plants and phytopathogenic fungi results in increasing proline accumulation, which promotes mycovirus transmission between incompatible fungi via inhibition of fungal non-self recognition system. Donor strain Ep-1PN is incompatible with recipient strain 1980. Source data are provided as a Source Data file.

phytopathogenic fungi, having significant implications for mycovirus ecology and holding practical importance for their application in biological control. Therefore, there is considerable interest in elucidating the mechanisms that underpin the plant-mediated weakening of VIC barriers.

In this study, we have observed an increase in plant-derived proline concentration in response to infections by phytopathogenic fungi. Elevated proline levels have been found to decrease fungal non-self recognition reactions, thereby facilitating enhanced mycovirus transmission between otherwise incompatible fungal strains and explaining the widespread mycovirus distribution in fungi. To the best of our knowledge, this data represents the first example of a plant-associated factor shown to promote horizontal mycovirus transmission and points to a novel plant defense mechanism against phytopathogenic fungi. Previous studies have indicated that higher plants exhibit elevated proline content in response to various environmental stresses[54]. Proline, acting as a signaling molecule, modulates mitochondrial activities, influences cell proliferation and death, and regulates gene expression, essential for the plant's recovery from stress[52]. Our data have shown that the addition of proline alone did not confer a

protective effect against *S. sclerotiorum* infection neither in laboratory nor field experiments, suggesting that proline primarily decreases disease severity by enhancing HAVs transmission through suppression of fungal non-self recognition reactions. Therefore, the plant's ability to diminish VIC reactions through proline production presents a novel potential strategy and mechanism against fungal infection in agriculture. However, the effect induced by plant proline in response to fungal infection is likely localized and sub-optimal, as the application of exogenous proline together with a HAVs-infected strain provides further protection to susceptible crops against phytopathogenic fungi. This treatment not only decreases disease severity, but more importantly, increases crop yields.

We have provided compelling evidence to support the notion that proline alleviates non-self recognition reaction in *S. sclerotiorum*. Firstly, proline treatment promotes hyphal fusion at the cellular level and reduces fungal ROS levels at the biochemical level. Secondly, proline inhibits the expression of genes encoding HET domain-containing proteins, and disruptions of these genes (Supplementary Figs. 9 and 10), including *Ssvic, Ssvib1*, and *Sscwr*, significantly promote mycovirus transmission. Previous studies have shown that homologs

of *S. sclerotiorum* genes downregulated by proline are essential for cell fusion and non-self recognition in *C. parasitica*, *N. crassa*, *B. cinerea*, and *P. anserina*[9,11,22,23,55]. Additionally, deletions of VIC-related genes (*vic1a-2*, *vic3b-1*, and *pix6-2*) in *C. parasitica* significantly promote CHV1 transmission between incompatible strains[23,56]. It would be interesting to further investigate how proline affects the expression of genes that regulate hyphal fusion in *S. sclerotiorum*.

Horizontal mycovirus transmission potentially occurs among fungi sharing the same ecological niche and is influenced by interactions between fungi and plants. Multiple mycoviruses have been observed to infect individual fungal strains at a high rate in nature[57]. In our study, we have discovered a significant increase in proline concentration in plants (oilseed rape, *Arabidopsis*, etc.) upon infection with a phytopathogenic fungus (*S. sclerotiorum*). This increase assists mycoviruses to circumvent the fungal non-self recognition reaction, potentially resulting in the mycovirus co-infections of individual fungal strains observed in nature. Furthermore, certain mycoviruses, such as Botrytis virus F or Sclerotinia sclerotiorum botybirnavirus 1, have been identified to infect fungi[24,25,30,58], including *B. cinerea*, *S. sclerotiorum*, and *Leptosphaeria biglobosa*, that occupy similar ecological niches in oilseed rape plants. Interestingly, a number of mycoviruses are found in different fungal genera, or even kingdoms[30,59,60]. However, whether proline can also promote hyphal fusion between different fungal species or genera remains to be determined in the future.

HAVs have potential as biological control agents but questions regarding their ability to spread in the natural populations of the pathogen and their ecological fitness under field conditions have to be addressed to expand their application in disease management strategies. Despite hindering fungal growth and rendering their hosts less ecologically fit than their virus-free counterparts, HAVs continue to spread in nature. For instance, CHV1 clearly impedes growth and sporulation, yet is naturally disseminated in European populations of *C. parasitica*[20,21]. Furthermore, some mycoviruses exhibit poor vertical transmission rates to conidia coupled with restrictions imposed by fungal non-self-recognition systems[43,61,62], but they endure in the natural fungal populations over time. Recent research has shown that HAVs can induce a change in the host fungus life style. For instance, SsHADV1 and Pestalotiopsis theae chrysovirus-1 can convert phytopathogenic fungi into beneficial endophytic fungi, promoting plant health[63,64]. Curvularia thermal tolerance virus infects an endophytic fungus and confers thermal tolerance to the plant[18], benefiting the survival of all three organisms under high temperatures (up to 65 °C) that would not be able to tolerate on their own. Here, we propose an additional explanation for the ecological stability of mycoviruses: plant proline promotes horizontal transmission of HAVs, continuously leading to new infections and compensating for the lower ecological fitness of the host. This tri-interaction system involving plants, phytopathogenic fungi, and HAVs, benefits plant health, survival of hypovirulent fungi, and mycovirus transmission. From this perspective, mycoviruses emerge as key players in the evolution and balance of local plant ecosystems, functioning similarly to viruses in marine ecosystems.

In addition to biocontrol applications in agriculture, our research provides a promising avenue for enhancing CHV1-mediated control of chestnut blight through the application of exogenous proline. Despite the potential of CHV1-based biological control, its success has been limited, particularly in North America. A previous explanation for this limited success is the complex structure of VCGs among *C. parasitica* strains in North America, which restricts CHV1 horizontal transmission. Previous studies have shown that *C. parasitica* infection increases proline production in chestnuts[65], and that CHV1 transmission rates are higher *in planta* than in vitro[21,38,66–68]. Therefore, we hypothesize that the enhanced transmission of CHV1

in chestnuts is mediated by plant proline, similarly to transmission of HAVs in *S. sclerotiorum*. In our research, plant proline levels induced by *S. sclerotiorum* are approximately 10-fold lower than the exogenous proline added in PDA and field experiments. Mycovirus transmission rate is significantly higher on PDA containing 3.2 mM proline than on plant leaves without exogenous proline. Addition of exogenous proline (3.2 mM) along with HAVs-infected Ep-1PN led to better biological control efficiency than Ep-1PN alone in our field experiments. Therefore, increasing the concentration of proline above the level produced by the plant in response to infection, through addition of exogenous proline, leads to a greater enhancement of mycovirus transmission in *S. sclerotiorum*.

In summary, fungi have evolved intricate non-self recognition mechanisms to prevent the spread of molecular parasites such as mycoviruses. In response to infection by phytopathogenic fungi, plants utilize proline to hinder fungal non-self recognition reactions, thereby reducing fungal resistance to mycoviruses (Fig. 5C). Our research elucidates the mechanism by which mycovirus survival and transmission can be enhanced in natural ecosystems and highlights a straightforward practical enhancement in the application of mycoviruses for fungal disease control under field conditions.

## Methods

### Strains and culture conditions
Information on *S. sclerotiorum* strains is summarized in Supplementary Table 1. All strains were routinely cultured on PDA at 20–22 °C to observe morphology and determine growth rate. Mycelial plugs were stored at 4 °C and dried sclerotia at −20 °C. The pathogenicity of all strains was determined by inoculating leaves of oilseed rape and *A. thaliana* with mycelial plugs.

### Plants and growth conditions
A commercial cultivar of oilseed rape (*Brassica napus* cv. Huashuang No. 4) was grown in pots with garden soil at 20 °C. *A. thaliana* (L.) Heynh., accession Columbia-0 (Col-0), was used as the wildtype. The T-DNA insertion lines p5cs2-1 (SALK_203144) in the Col-0 background were obtained from Arashare (https://www.arashare.cn/), and confirmed using PCR with three primer pairs (Supplementary Fig. 12 and Supplementary Table 2). After stratification at 4 °C in the dark for 2–4 days, *A. thaliana* seeds were planted and grown in soil at 20–22 °C, with 50% relative humidity and a 16-h light/8-h dark photoperiod for 4–5 weeks.

### Horizontal transmission assays
HAV-infected hypovirulent donor strains (Supplementary Table 1) were co-cultured with HAV-free virulent recipient strains, labeled by a hygromycin (1980R or 1980m) resistance gene, in vitro (PDA or proline-containing PDA) or *in planta* (oilseed rape or *A. thaliana*). Following contact of donor and recipient strains for 3–5 days in vitro or 24–48 h *in planta*, the recipient strain was re-isolated for sub-culturing on PDA containing 50 ng/μl hygromycin or 50 ng/μl cephalosporin. RT-PCR with mycovirus-specific primers was used to establish infection status in recipient strains and quantify horizontal transmission rates (Supplementary Table 2).

### RNA extraction, RT-PCR, and RNA-seq analysis
*S. sclerotiorum* strains were cultured alone or co-cultured with (in)compatible strain on: (1) cellophane membrane overlaying PDA with or without proline; (2) oilseed rape or *Arabidopsis* plants. Subsequently, mycelia and plant issue were, respectively, collected and total RNA was extracted using a commercial RNA extraction kit (Takara, Code No.: 9108). RNA quantity and quality were assessed using an Agilent 2100 Bioanalyzer (Agilent Technologies, CA, USA) and a NanoDrop spectrophotometer (Thermo Fisher Scientific, USA).

Total RNA (about 1 μg) was treated with RNase-free DNase I (Takara, Code No.: 2270A) and used to synthesize complementary (c) DNA with M-MLV reverse transcriptase (Takara, Code No.: 2641A) and oligo (dT) primers. Quantitative (q)PCR was performed using iTaq SYBR Green Supermix in a Bio-Rad CFX384 Real-Time PCR System (Bio-Rad). The actin gene was used as a control. All RT-qPCR primers are listed in Supplementary Table 2.

RNA-seq was performed using the HiSeq 2500 platform with at least three biological replicates (Illumina, USA). Approximately two million reads were obtained for each sample. Raw sequence reads were trimmed and clipped to remove low-quality reads and adapter sequences using Trimmomatic (version 0.9.25). Processed reads were mapped against the *S. sclerotiorum* reference genome (http://fungi. ensembl.org/Sclerotinia_sclerotiorum_1980_uf_70_gca_001857865/ Info/Index) using HISAT2 (version: 2.0)[69]. The aligned SAM files were sorted and converted into BAM files using SAMtools[70]. A count table was generated using featurecounts (version: 1.6.0)[71]. Differential gene expression analysis was performed using the R package DEseq2 and heatmaps were generated using R[72,73].

### Evans Blue and NBT staining
To detect PCD in *S. sclerotiorum*, Evans Blue staining was conducted as previously described[53]. Briefly, incompatible strains 1980m and Ep-1PNA367G were co-cultured on PDA with or without 3.2 mM proline. Upon contact, the strains allowed to interface for 12 h and then stained in a 0.5% (w/v) Evans blue solution for 30 min. The interface was observed under a stereomicroscope after being washed twice with phosphate buffer saline (PBS, pH = 7.4).

To detect ROS levels in *S. sclerotiorum*, NBT staining was performed as previously described, with minor modifications[53]. The incompatible strains 1980m and Ep-1PNA367G were cultured separately on PDA for 36 h. The mycelium of Ep-1PNA367G was collected, ground into powder in liquid nitrogen, and re-suspended in PBS. The supernatant was collected by centrifugation at $12,000 \times g$ for 5 min and 3.2 mM proline was added. This mixture was used to treat 1980m on PDA for 30 min. Then, 1980m was washed twice with PBS and stained with 0.5% (w/v) NBT solution for 30 min in the dark. The reaction was terminated with ethanol, and ROS levels were quantified colorimetrically. All biological experiments were repeated at least five times.

### Microscopy
The cassettes of green and mCherry fluorescent protein (GFP and mCherry) under the control of two *Aspergillus nidulans oliC* promoters are shown in Supplementary Fig. 11 (Genbank accession number: PP209073-PP209074). Strains 1980 and Ep-1PNA367 were labeled respectively with mCherry and GFP using a standard polyethylene glycol (PEG)-mediated protoplast transfection protocol as previously described[74]. The resulting strains were denoted as 1980m and Ep-1PNA367G. These labeled strains were co-cultured on a sterilized glass slide covered by a thin PDA layer, with or without proline, at 20 °C. Upon contact, an Olympus confocal microscope or TEM or SEM was used to observe the interface between the strains. The excitation and emission wavelengths were 552–580 and 395–488 nm for mCherry and GFP, respectively. All biological experiments were repeated at least five times.

### Generation of gene deletion mutants
*S. sclerotiorum* genes (Supplementary Table 2) were knocked out using the split-marker method, as shown in Supplementary Fig. 8. Briefly, segments approximately 1.2 kb in length located upstream and downstream of each gene were amplified from the *S. sclerotiorum* genome with specific primers (Supplementary Table 2). These segments were then ligated with two segments of NE and EO carrying a 0.6 kb overlapping region. NP and PT were obtained from

a neomycin-resistance gene cassette using the ClonExpress® II One Step Cloning Kit (Vazyme Code No.: C112-01). Equimolar amounts of the purified NP and PT fragments were used to transfect *S. sclerotiorum* protoplasts using a standard PEG-mediated protocol as previously described[75]. Transfectants were transferred onto PDA containing 100 ng/μl hygromycin or neomycin for continuous cultivation and screening until stable phenotypes were observed. Initially obtained transfectants were typically heterozygous since *S. sclerotiorum* cells are multinucleate. Therefore, multiple rounds of single-protoplast isolation for each candidate transfectant were conducted following a previously reported protocol with minor modifications[75]. Fresh hyphae of candidate transfectants were incubated at 28 °C in 1 ml cell wall digestive solution (1% megalyase in 0.7 M NaCl solution) for 10 min under shaking ($140 \times g$), and the mixture was filtered through paper to remove mycelial fragments. The filtrate then was centrifuged at $5000 \times g$ for 5 min, and the precipitate was resuspended in 600 μL STC (1 M sorbitol, 50 mM Tris-HCl pH = 8.0, 50 mM CaCl$_2$) and spread on regenerative medium containing hygromycin or neomycin. Subsequently, each single-protoplast transfectant was confirmed to be homozygous by multiple PCR amplifications as shown in Supplementary Fig. 10.

### Liquid chromatography-mass spectrometry (LC-MS)
*S. sclerotiorum* strains were inoculated on leaves of *B. napus* (2 months old) or *A. thaliana* (5–6 weeks old). The leaves were collected at 12 hpi, freeze-dried and crushed using a mixer mill (MM 400, Retsch) with zirconia beads for 1.5 min at 30 Hz. Proline was extracted overnight at 4 °C from 100 mg leaf powder with 1.0 ml 70% (v/v) aqueous methanol containing 0.1 mg/l lidocaine as an internal standard. Following centrifugation at $10,000 \times g$ for 10 min, 0.4 ml of each lipid-soluble extract was mixed and filtered (SCAA-104, 0.22-μm pore size; ANPEL, Shanghai, China, www.anpel.com.cn) before LC-MS analysis. LC-MS/MS was performed by coupling a Dionex Ultimate 3000 ultra-performance liquid chromatography system (Thermo Fisher Scientific) with a TSQ Altis™ triple-quadrupole mass spectrometer (Thermo Fisher Scientific).

### Field experiments
Experiments were conducted on oilseed rape fields in Hubei province, in Huanggang for 2 years (2021 and 2022) and in Wuhan for 1 year (2022). Plants were regularly treated with fertilizer, but not fungicide, during the growing season. Twelve plots of approximately 50 m² (5 m × 10 m width × length) were defined in each field and each treatment was applied to at least three plots. Hyphal fragment suspensions of Ep-1PN were prepared as previously described[63]. Hyphal fragment suspensions with or without 3.2 mM proline were sprayed on aerial plant parts at the flowering stage (earlier than the epidemic phase of stem rot), and 3.2 mM proline or water instead of hyphal fragment suspension was used as control. 100 plants were randomly chosen from each plot, and the disease severity for each was rated on a scale of 0 to 4[76]. Plants within 20 m² areas were harvested from each plot, and their seeds were weighed to calculate the disease index. All data were subjected to statistical analysis.

### Quantification and statistical analysis
The quantitative data for mycovirus transmission rates, proline content, and fungal virulence assays are presented as mean ± standard deviation (SD). The statistical analyses were performed by Student's *t* test analysis of variance, Figures were generated by the GraphPad Prism 8.0 software.

## Data availability
The RNA-seq data generated in this study have been deposited in the NCBI database under accession code (Bioproject Accession: PRJNA908905; PRJNA1065649; PRJNA1068540). All of our raw data

including full uncropped images in the manuscript are provided in figshare (https://doi.org/10.6084/m9.figshare.25108211) Source data are provided with this paper.

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

## Acknowledgements

This research was supported by the National Natural Science Foundation of China (32072475 and 32130087) (J.X.), the National Key Research and Development Program of China (2022YFA1304400) (D.J.), the Fundamental Research Funds for the Central Universities (2021ZKPY005 and 2662023PY006) (J.X.), and the Earmarked Fund for CARS-12 (D.J.). We would like to thank Professor Kenichi Tsuda and Professor Xiaowei Han at Huazhong Agricultural University, and Professor Nemat O Keyhani at the University of Florida for their constructive suggestions and proof-reading of the manuscript. We thank the National Key Laboratory of Agricultural Microbiology Core Facility for assistance in microscopy observation.

## Author contributions

D.H., J.X., and D.J. conceived the project. D.H. executed experiments, analyzed data, and prepared the manuscript. J.L. designed and performed the gene knockout experiment. H.Y. and W.C. conceived the LC-MS experiment. J.C., Y.F., X.X., Y.L., T.C., B.L., X.Y., and Q.C. carried out data collection, analysis, and interpretation. I.K. participated in the guidance of manuscript writing and provided comments for the manuscript. All of the authors discussed the results and commented on the manuscript.

## Competing interests

The authors declare no competing interests.
