## [Peer Review File · Nature Communications]

Plants interfere with non-self recognition of a
phytopathogenic fungus via proline accumulation to facilitate
mycovirus transmissionREVIEWER COMMENTS

Reviewer #1 (Remarks to the Author):

The work submitted by Hai et al. focuses on a topic of considerable interest. Mycoviruses (fungal viruses) have gathered great attention as potential biocontrol agents of fungal diseases since the discovery of CHV1, a mycovirus that induces strong hypovirulence in its fungal host, *Cryphonectria parasitica*, and had spread naturally in the population of this pathogen in Europe. However, mycoviruses generally lack an extracellular route of transmission and they spread only by intracellular mechanisms. Fusion of vegetative hyphae is the major route of mycovirus horizontal transmission. Therefore, a major concern in the practical application of myoviruses for the control of fungal diseases is that the complex vegetative compatibility/incompatibility systems developed by filamentous fungi to limit hyphal fusion will impair the spread of the mycoviruses introduced in the environment as biological control agents. Some previous data and observations suggested that environmental factors might promote mycovirus transmission even between incompatible strains. The results presented here provide compelling evidence that the plant environment does indeed weakens vegetative incompatibility barriers greatly enhancing mycovirus transmission. Furthermore, the mechanism by which the plant weakens vegetative incompatibility promoting mycovirus spread is elucidated. I consider these results of considerable interest as the success of mycoviruses as biocontrol agents is strongly dependent on their ability to spread in the population of the pathogen in natural conditions. What is more, these results can have practical application in the implementation of mycovirus-based strategies to control fungal diseases. The work is very complete as they have approached the topic from a number of angles, generating data that combined strongly support their findings.

However, in spite of the merits of the research presented, the manuscript has many problems that need to be addressed. The text conveys a lack of attention to how things are expressed making it difficult to follow; one requires a high degree of previous knowledge on the topic to understand the relevance of the work, and even how many experiments were performed and therefore to properly evaluate the results. I have the strong feeling reading the current version of the manuscript that only readers very familiar with this type of research would be able to properly understand the work. Besides, the text conveys a lack of clarity of concepts and some of the information provided is inaccurate or wrong. I also have some concerns regarding some experiments and the quality of some images that I will refer to in the corresponding section.

Below I explain in more detail my concerns and provide comments and suggestion regarding each section of the manuscript.

INTRODUCTION

The authors could do a better job in the Introduction describing the role that fusion of vegetative hyphae has in the spread of mycoviruses. This would provide the necessary context to understand the relevance of their results that can be of great interest in positively re-assaying the potential of hypovirulence-inducing mycoviruses (HIVs) as biological control agents against fungal diseases. Information regarding the role of vegetative hyphae fusion as a major route of mycovirus horizontal transmission is dropped here and there but in a rather disorganized manner. As a matter of fact, some of the information included in the Introduction is not accurate. For example:

Lines 40-41: The fusion of vegetative compatible hyphae does not generally result in fusion of both nuclei generating a diploid nucleus, as stated, and to refer to this process as "mating" creates confusion. Sexual reproduction and the parasexual cycle of exchange of genetic material through the

fusion of vegetative compatible hyphae are two distinct processes even though it can be some overlap in the molecular mechanisms that regulate them. Hyphal anastomosis is the major route of horizontal mycovirus transmission. Sexual reproduction, in those filamentous fungi with sexual cycle, doesn't seem to have a major role in the transmission of mycoviruses between strains. CHV1 for instances is not transmitted to sexual spores. They have to explain these things in a clear manner providing the necessary context for the research presented.

Lines 59-60: The authors state that mycovirus can be acquired via infection by viral particles and then transmitted by intracellular mechanisms. That's inaccurate, extracellular routes of transmission are not prevalent in mycoviruses, in most cases transmission only takes place through intracellular mechanisms like hyphal anastomosis (horizontal transmission) and sporulation (vertical transmission). Funnily enough, to support their statement that initial infection can take place through viral particles they cite a review article on CHV1, a capsidless mycovirus that doesn't therefore generate viral particles. The authors indeed admit that transmission through intracellular mechanisms is the key route of infection later on in the Discussion, when they provide the weakening of vegetative incompatibility mediated by proline generated by the plant as a putative explanation of the identification of the same mycovirus in different fungal species. If one thinks about it, this distinction between first acquired infection (that can be via viral particles) and then transmission (that takes place through hyphal anastomosis) makes no sense; when a mycovirus is transmitted by hyphal anastomosis the recipient strain is "acquiring infection". Thus, this statement conveys lack of clarity of ideas.

The rest of this paragraph is also confusing; let's check for example this sentence "Despite the fact that VIC groups are extremely complex in filamentous fungi, identical mycoviruses have been frequently characterized in fungal isolates found in nature and belonging to different genera or species." First of all, what does it mean that a VIC group is extremely complex? I guess that what the authors mean is that filamentous fungi can have many different VIC groups, that is, it's the structure of VIC groups what is complex. To make this point clearer they could mention that within a fungal species, strains that harbor the same alleles in the genes that control this process are compatible to undergo hyphal anastomosis and are placed in the same VCG (vegetative compatibility group) whereas those that have different alleles are vegetatively incompatible and belongs to different VCGs. And then explain that the genetic control of hyphal anastomosis can be very complex resulting in very complex VCG structures with a high degree of vegetative incompatibility within the fungal species. In this way the concept of complexity regarding VIC can be properly conveyed, a concept that is quite relevant to the research presented here. There is another problem with this sentence, the fact that in a fungus the VCG structure is very complex (first part of the statement) affects how mycoviruses are spread within a fungal species not their transmission to other fungal species (second part of the statement). Therefore, this sentence should be split in two, first stating that in some filamentous fungi mycovirus infection is very prevalent even though their VCG structure is very complex, suggesting that there are mechanisms in natural conditions that weaken vegetative incompatibility. Secondly, the identification of the same mycoviruses in different fungi further reinforces this idea suggesting that incompatibility can be weakened in nature to the point of sometimes allowing horizontal transmission even between different fungal species.

These points are not minor, they convey a lack of clarity of concepts that is very detrimental to the manuscript.

The broad presence of mycoviruses in filamentous fungi with very complex VCG structures and the identification of the same mycoviruses as naturally infecting different fungal hosts, are observations that suggest that there are environmental factors that weaken vegetative incompatibility promoting mycovirus transmission. Besides, there are previous works suggesting that horizontal mycovirus

transmission might be enhanced in the plant environment. The authors only cite them at the end of the Discussion; I consider those paper (Brusini et al. 2013 and Suzuki et al. 2021) should already be cited in the Introduction. The Introduction should clearly provide the context (and rationale) for the research presented here, that is: (1) The potential of mycoviruses that induced hypovirulence as biological control agents is strongly dependent on their ability to spread in the natural populations of the pathogen, (2) hyphal anastomosis of vegetative hyphae is the major route of mycovirus transmission, (3) the complex vegetative incompatibility systems developed by fungi is therefore a concern in the potential application of mycovirus-based strategies to disease control, (4) however, some observation and previous results suggested that mycovirus spread in natural conditions is more efficient than predicted by the high degree of vegetative incompatibility that might be present in fungal populations.

I'd also mention in the Introduction that in the implementation of mycovirus-based control strategies mycovirus are introduced in the field through infected strains. The first time this appears in the manuscript is when they described their field experiment. This kind of key information helps readers that are not experts in this topic to most easily follow the work.

RESULTS

Section "Plant-fungus interactions enhance mycovirus transmission":

Line 95: After explaining that mycoviruses are readily transmitted to a compatible strain but not to incompatible strains on PDA, they refer to Fig. 1B. However, Fig. 1B illustrates transference of HIVs (hypovirulence-inducing mycoviruses) not on PDA but when strains are co-inoculated in planta, and it shows successful transmission between incompatible isolates. So, it is misleading to refer to Fig. 1B at this point. Also, the authors should mention in the text key information to understand the results on PDA which are illustrated in Fig. 1, that is that successful transmission on plates is illustrated by the recipient strain acquiring the abnormal phenotype of the virus donor strain.

Line 99: Delete reference to Fig. 1C at this point; just keep references to Fig. 1B, 1C and 1D in line 101.

Lines 103: They should direct the reader at this point to Supplementary Fig. 1; the doubts I have about how the experiment was performed were answered after checking Supplementary Fig. 1 later on when they refer to it in line 108.

Regarding Supplementary Fig. 1 I'd change a few things to make it clearer. To place Dual culture on the arrows is misleading. The co-inoculations are before the arrow and the arrows better illustrate the selection of the recipient strain on PDA + hyg after the co-inoculation of both strains. This sentence in the legend is very confusing: "The isolates were re-isolated from strain 1980m that was dual-cultured with Ep-1PN, and subcultured on PDA containing hygromycin..." What they do after co-cultivation is to specifically select the virus recipient strain by growing on PDA + hyg, as only the recipient strain harbor a hygR marker; this should be clear in the text.

I'd like to emphasize again that the authors tend to described the experiments performed in such a way that only researchers that do this type of assays themselves would be able to understand them.

Regarding specifically Supplementary Fig. 1, changes should be made to both the figure and the figure legend to better convey what they did, that is: (1) co-inoculate both the virus donor strain and an incompatible virus-free strain that had been labeled with a hygR marker, (2) re-isolate the fungus from plant (or from PDA) specifically selecting the recipient strain by growing on PDA + hygR, (3)

analyze presence of the mycoviruses in the isolated recipient strains. It seems they first grow on PDA + hyg to specifically select the recipient strain and then transfer to PDA to assess phenotype but I'm unclear about that.

I see different phenotypes on plate in Supplementary Fig. 1; it would help if the authors explain if single infections of each mycovirus induce alterations in growth and described them, as they did with the doubly infected strain.

It took me some time to understand why the RNA-seq results were split in two figures, Fig. 2 and Supplementary Fig. 2, until I noticed that in the results presented in Fig. 2 they combined the HIV-infected strain with an incompatible virus-free strain whereas in the results presented in Supplementary Fig. 2 both strains were uninfected (in this case they used the uninfected version of the HIV-infected strain). This is of interest, I assume they did that to determine if the HIVs themselves might have an effect on gene expression associated to VIC but nothing of that is explained or discussed in the text; the experimental set ups and the results obtained have to be clearly explained so they can be properly evaluated.

There are inconsistencies in the figures that might look minor but create confusion. For example, in Fig. 2 the panels say PDA whereas in Supplementary Fig. 2 it says Medium.

Also, I miss the inclusion of a compatible interaction as a control.

Section "Plant proline accumulation induced by fungal infection enhances mycovirus transmissibility":

Line 131: After and post-inoculation don't go well together as post means after. In fact, I'd change the sentence to just "..., was upregulated in both oilseed rape and *A. thaliana* Col-O infected with virulent virus-free strain Ep-1PNA367 (Supplementary Fig. 4A)" as the figure shows that the increase of expression starts at 7 hours but the time course is quite different in oilseed rape and *A. thaliana*.

Lines 133-135: The hypothesis the authors outline here is not properly understood if they don't start by saying that vegetative incompatibility has been associated to the induction of PCD that prevents the formation of viable hyphal bridges between both strains. There is another problem, the work illustrated in Fig. 3 does not test any hypothesis regarding inhibition of non-self recognition or PCD what it does is to confirm that the increased mycovirus transmission rates observed in planta are indeed mediated by proline produced by the plant. So, the hypothesis they are really testing here is that the enhanced mycovirus transmission observed in planta is mediated by proline produced by the plant and this is what they should state at this point.

Line 135-137: Very confusing sentence, I propose to change it to: "To test this hypothesis, we obtained an *A. thaliana* Col-0 mutant with a T-DNA insertion in p5cs2 which was impaired in proline production during *S. sclerotiorum* infection (Fig. 3B)"

Line 140: I guess that what the authors mean by "but did not enhance resistance (Fig. 3E)" is that addition of proline in the inoculation of oilseed rape with the virulent strain did not induce any protection against infection. They should express this idea in a clearer way as it is important, it provides additional evidence that the effect of proline is associated to increased mycovirus transmission and not the induction of resistance in the plant as the effect of proline in decreasing disease severity is only observed when the HIV-infected strain is present.

Line 141: Change "detected" to "tested for the presence of". I take the opportunity here to note that there are other instances in the text where the terms used sound odd, and emphasize again that the

entire manuscript should be thoroughly revised. The text requires too much revision for me to be able to suggest specific changes throughout.

Lines 144-150: This paragraph is very confusing, it should be re-written. It is difficult to understand what is the point they're trying to make. Looking at the figure what I see is confirmation in in vitro culture that proline enhances mycovirus transmission and this exactly what they should highlight in the text.

Lines 151-155: This paragraph is rather messy and therefore difficult to follow. I'd express things in a simpler manner along these lines:

"To test if the proline-mediated promotion of mycovirus transmissibility in *S. sclerotiorum* is a general phenomenon, we measured horizontal transmission rates of HIVs in five previously reported hypovirulent strains, SZ-150, SX276, Ep-1PN, WF-1, SCH733 and AH98 (Supplementary table 1 and Fig. 3I), to incompatible strains SG9 and 1980m. For that, each HIV-infected donor strain was co-cultured with virus-free recipient strains SG9 and 1980m, on PDA and on PDA plus 3.2 mM proline."

Section: "Proline promotes hyphal fusion, inhibits expression of vic-related genes, and reduces ROS production"

This is the section that can be considered to focus on testing the hypothesis stated in lines 133-135, so that information should be brought here. And like I said, it is important to mention that VIC has been associated to induction of PCD. In this section is where they really assay if proline has an effect on hyphal anastomosis.

Regarding the title of this section, I find it somehow misleading. It seems to be claimed that any gene that plays a role in vegetative compatibility/incompatibility is a vic gene. This is not the case, vic genes are better described as those specifically associated to the process, the genetic determinants of the structure of VCGs (Vegetative Compatibility Groups) of a fungal species. In this section the authors perform a general study of changes in transcription patterns in response to proline. For example, G-protein signaling might play a key role in controlling hyphal fusion but that doesn't mean that the components of G-protein signaling pathways are vic genes. The same goes for the title of Fig. 4.

The inclusion of a compatible interaction in this study would've allowed to compare gene expression in the incompatible interaction in the presence of proline with gene expression in the compatible interaction.

Line 165: They mention strain SCH733 but none of the results presented in either Fig. 4 or Supplementary Fig. 6 include this strain. The authors don't explain what the arrows points at in Supplementary Fig 6; I can see hyphal degradation in the upper panel of this figure but I don't know what the arrows point at in the lower panel.

Lines 172-174: Fig. 4C is poor quality and to be honest I don't know what I'm supposed to look at in this figure. In that sense, the legend is little helpful, it states:

"PI staining and visualization using confocal microscopy for detection of cell fusion or cell death at the interface between 1980m and Ep-1PNA367G."

What cell fusion are we seeing here? And again, they don't explain what the arrows mean. According to the M&M strain Ep-1PNA367G is the one labeled with GFP, so what cell fusion do we see under GFP fluorescence? In the presence of proline all the structures observed in the DIC image are also observed in the GFP image. All I can say in this figure is that there is no staining with PI in the presence of proline pointing to lack of cell death.

Lines 174-176: Fig. 4D is really poor quality. If the authors want to be able to illustrate viable hyphal anastomosis between incompatible strains in the presence of proline with the two strains labelled with different fluorescent proteins, they really need to show better images. Also, the legend should be more informative. And like I said, it would add to the work to illustrate how the compatible interaction looks at the microscopic level and compared with the incompatible interaction plus proline.

Line 178: It's not the gene expression profile of *S. sclerotium* but that of the interaction of two vegetative incompatible *S. sclerotium* strains. I emphasize again that more care should be taken to explain the work properly.

Lines 186-188: I assume the RNA-seq data in planta they compare here with the response to proline on PDA, is the same one presented in Fig 2 (first section of the results); they should make this clear. Besides, the references here to Supplementary Fig. 4B is confusing.

Line 193: Checking the Supplementary figure, I see that what the authors did was targeted deletion of the entire gene ORF, so it'd be more appropriated to say that they generated deletion mutants. And that all throughout the text. For example, in line 196 they should say "Ssvc2 deletion resulted in...."

Lines 200-203: Odd and confusing sentences. First, I think that the second sentence should go first, and then state that the highest increase in mycovirus transmission was observed in the mutant deleted for Ssvc3. Besides, the way the experiment was performed need to be clearly explained; I guess they co-cultivated the two strains and then specifically selected the recipient strains growing in the presence of neomycin, as the mutants harbor a neoR marker, and then they tested for the presence of the HIVs in the recipient strains. They say in the legend that N=24 for the co-cultures; does that mean they analyze 24 samples per pair of donor and recipient strain? I'm guessing all this for what I know about this type of assays, for somebody with no experience in this type of experiments it would be even more difficult to understand.

There is another point to be made. The authors keep saying that strains where converted to hypovirulent when describing results where it is clear that they didn't do any virulence assay, and that's confusing. It would be clearer to say that strains exhibited the phenotypes associated to hypovirulence. They could also explain at some point early in the text that as infection by the HIVs under study induces clear alteration in the growth of the host fungus, these phenotypic alterations can be used as a visual marker of acquired infection.

Line 203: There is no Fig. 4I, I think they mean Fig. 4J.

DISCUSSION

More attention should be paid to the structure of the Discussion, it looks somehow disorganized.

Line 217: I consider that the information provided in the second paragraph (lines 226-238) should be included at this point. This is the place to mention previous data suggesting that the plant could improve mycovirus transmission and how it is of considerable interest to study this further, to determine if it's a general phenomenon, and to elucidate by which mechanism the plant induces weakening of the incompatibility barriers.

Line 220: The statement "thus acting as a mechanism against fungal infection" could be a separate sentence explaining that as vegetative incompatibility is thought to have evolved to limit the spread of extrachromosomal genetic elements detrimental to the fungus such as mycoviruses and senescence

plasmids, the weakening of vegetative incompatibility reactions by the plant can act as a mechanism against fungal infection.

Lines 220-225: The production of proline by the plant seems to be a response to infection. It would help if the authors provide more information on what is known about the role of plant proline during infection. It is true though that in the field assays, adding proline alone didn't have a protective effect against infection. It seems clear that the main effect on proline in decreasing disease severity was by enhancing HIVs transmission.

The paragraph between lines 239 and 257 mixes information on different topics. The discussion about how zinc compounds were previously shown to weaken VIC but this is the first report of a plant-derived compound promoting hyphal anastomosis could be a separate paragraph. As for the information regarding identification of the mycoviruses in different fungal species, they can discuss this later, mentioning that it would be of interest to determine if proline can also promote hyphal anastomosis between different species.

Lines 258-276: This part of the discussion is too speculative and sometimes difficult to follow. It is indeed of interest to discuss the case of CHV1 which is used for disease control in Europe but doesn't work well in North America, and how the results presented here provide an avenue to explore to try and improve the potential of CHV1 as a biocontrol agent. However, I suggest that this part of the discussion is re-written to increase clarity and avoid too much speculation. What the authors can safely claim is that even if the production of proline by the plant and its promotion of hypovirus transmission is a general mechanism, their data show that increasing the concentration of proline above the level produced by the plant in response to infection, by addition of exogenous proline, has a greater impact on hypovirus transmission. Therefore, they postulate that applying the appropriate amount of proline together with the CHV1-infected strains could significantly increase CHV1 transmission rate and it is something worth exploring.

Lines 277-285: This paragraph is confusing. It must be a good idea to enumerate the different studies they performed that combined provide compelling evidence that proline alleviates non-self recognition in *S. sclerotium*, but I feel such information should be included earlier in the Discussion. On the other hand, if they mean to focus this paragraph on the genetic determinants of vegetative compatibility/incompatibility, they need to re-write it to better conveyed the point they want to make. The authors appear to argue that because the genetic regulation of vegetative compatibility/incompatibility is similar in different fungi, they have shown that proline is a general regulator of this process. They have no grounds to make such statement. At most they can argue that it would be interesting to investigate how proline affects the expression of genes that regulate hyphal anastomosis in fungi where the process is well studied and those genes have been identified. I guess they based their claim on the results presented in Supplementary Fig. 3 but those results only show that G-protein signaling is downregulated in fungi during infection. G-protein mediated signaling pathways play many roles in fungi, including roles in pathogenicity.

I should also mention that as in other parts of the manuscript, the information the authors provide about the genetic regulation of vegetative compatibility/incompatibility is not accurate. There are not a set of homologous genes governing VIC in filamentous fungi. The situation is more complex, genes regulating vegetative compatibility/incompatibility can be quite diverse in the different fungi. Even within a fungal species, the genes whose allelic differences determine vegetative compatibility/incompatibility are very diverse, have different molecular function and some of them have a HET domain and some don't. This is why identification of these genes is not straight forwards. A fungal species in which the genetic of VCGs is well characterized and illustrate the points I'm making is *C. parasitica* (see for example, Choi et al. 2012, *Genetics* 190: 113-127 and Zhang et al. 2014,

Genetics 197: 701-714).

Lines 286-295: I think this point is quite interesting but it is broader than stated by the authors and should be discussed in more detail. There is the long-standing question why mycoviruses that negatively affect fungal growth are still spread in nature, they are not lost even though they are expected to be less fit ecologically than their virus-free counterparts. CHV1 for example has a clear negative impact on growth and sporulation and yet it had spread naturally in the populations of *C. parasitica* in Europe. Some mycoviruses don't have a good rate of vertical transmission to conidia and yet they're present in the natural populations of the fungus and are not lost over time. I think it is interesting to discuss this question in light of the results presented in this manuscript. The plant promoting horizontal transmission of these hypoviruses resulting in continuous new infections could make up for the lower ecological fitness of the strains harboring them.

Lines 295-297: These statements make little sense and should be deleted or re-phrased so that the point the authors are trying to make here can be understood.

Line 301: Delete reference to Fig. 5C, this is a general statement regarding the results provided here not something specific of Fig. 5C.

MATERIALS AND METHODS

Line 311: By inoculating leaves with what? Looking at the pictures of oilseed rape I guess they use plugs of mycelia.

Lines 317-318: The expression "genomic polymerase chain reaction (PCR)" sounds strange; I guess what they mean is PCR using plant genomic DNA as a template. Like I said before, the authors need to fix these problems that makes the text look like informal talk.

Line 323: Section "Mycovirus horizontal transmission assay." There is an abuse of information in parenthesis in this section.

Line 330-331: I guess it'd be more accurate to say that "Reverse transcription (RT) followed by PCR with mycovirus-specific primers was used to assay infection in the recipient strains to quantify the rate of mycovirus horizontal transmission (Supplementary table 2)." Also, the Supplementary Table 2 has to be fixed, terms like "Detection for" and "PCR for upstream fragment of" sound very odd.

Line 332: Section "Nucleic acid extraction, RT-PCR and RNA-seq analysis." They say nucleic acid but they only talk about RNA, so they should say RNA extraction.

Lines 33-334: The first sentence of this section is very confusing; it is clear that they want to include in just one sentence all the different biological materials they use for RT-PCR and RNA-seq but that doesn't work. Either they split this information in several sentences so that it makes sense or just say that they extracted RNA from both plants and *S. clerotiorum* grown in vitro using a commercial.... I favor the first option.

Line 368 (Section Microscopy): Which GFP and mCherry cassettes did they use for the expression of GFP and mCherry? To provide that information doesn't take much room and makes the experiments easier to understand. Like I said, there is seemingly little care about how things are expressed throughout the text and in this section, there is a little example of that, the authors claim that the transformation resulted in fluorescent strains such and such; it would be more accurate to say that they named the resulting strains such and such.

Line 369: It should be specified "PEG-mediated protoplast transformation."

Line 374: Section "Generation of gene KO mutants." Like I said before, they should talk about gene deletion mutants rather than KO mutants. Besides, whereas in most sections of the Materials and Methods they give little information, in this one they seem to provide too many details. This section could be shortened, for example from line 381 they could say "Equimolar amounts of the of the purified NP and PT were used to transform protoplasts of *S. sclerotiorum* using a standard PEG-mediated protocol." And then explain how transformants were selected. The last sentence "and were purified by single-protoplast isolation to obtain homozygous strains." makes no sense to me; how single-protoplast isolation is going to be performed after you obtain your transformants?

Line 408: I guess that what the author mean is "earlier than the epidemic phase of stem rot" or "before the epidemic phase of stem rot"

FIGURE LEGENDS

Figure legends in general do a poor job describing the result presented in the figures; they should be thoroughly revised so that the figures can be understood without having to check other sections.

Reviewer #2 (Remarks to the Author):

The manuscript titled "plants interfere with non-self recognition of fungi via proline accumulation to facilitate mycovirus transmission" detailed evidence supporting an observation that identical mycoviruses frequently found in fungal isolates in nature and belonging to different genera of species. Their data support that mycoviruses are less restricted by vegetative compatibility when the fungal hosts co-infecting plants than in vitro without plants. The authors presented multiple gene differential expression data, proline accumulation, mycovirus transmission rate, TEM, ROS, knockout mutants for genes potentially associated with proline's sequestering activity on ROS upon anastomosis. Overall, this is a well-done and well-written manuscript.

One concern on the generality of this study described with their manuscript title, whether this claim applies to other fungi and mycovirus genotypes or can be demonstrated between *Sclerotinia sclerotiorum* and another fungal species in a cross-species manner. For example, whether SsDRV and SsRVL or the co-infecting SsEV1, SsHV1, SsNSRV1, SsMYRV4, and SsPV1, can be transmitted to another fungal species such as *Botrytis cinerea* in plants that are a common host to both pathogens. Some of these viruses may have been reported in a virome study of *B. cinerea*, and it would be important to know if a transmission can be artificially induced by co-infecting them on plants or proline-amended PDA. There had been reports between the two fungi, although this reviewer does not have time to look into the overlapping members of viromes.

This reviewer recommends the authors to either include such data to support the claim or instead, specify that this work tested dsRNA mycovirus transmission between different *S. sclerotiorum* species that don't typically form anastomosis on usual PDA. In the later case, the title will need to be modified accordingly.

Reviewer #3 (Remarks to the Author):

This original research describes a significant finding that deserves publication in this journal. The research is a breakthrough as it solves a puzzle as to where and how mycoviruses can transmit between fungal hosts that are themselves vegetatively incompatible. The research also provides novel avenues for application of their insights into field-based research. The research is sound methodologically with sufficient detail provided. The manuscript is well written and the figures are concisely presented. There are some minor edits suggested below.

Introduction

Consider citing Arshed et al. 2023 (DOI: 10.1016/j.fgb.2023.103827) who described VIC locus genes Bcvic1 and Bcvic2 in *Botrytis cinerea*, that is more closely related to *Sclerotinia sclerotiorum* than the other fungi whose VIC genes are referenced in the Introduction

Line 243. Do the authors mean “transmitted” or “present” in the sentence “Interestingly, a number of mycoviruses are transmitted between species, genera, or even kingdoms”

Line 269-270 “... also found that proline could inhibit expression of a series (series?) of genes related to fungal non-self recognition, and confirmed (confirmed?) via knock-out experiments...”

Line 272-276 Therefore, although proline (is) induced by *C. parasitica* infection to enhance mycovirus transmission between incompatible strains, the concentration of proline in chestnut or other plants maybe (may be) too lower (low) to largely enhance mycoviruses (mycovirus) transmission, and then (delete ‘then’) eventually fail to accomplish the desired control efficiency on chestnut blight in North America.”

Line 277-278 “We provided compelling evidence to (the notion) support that proline alleviates non-self recognition reaction between two incompatible fungal strains in *S. sclerotiorum*.”

Line 299-301 “In the process of combatting infection of pathogenic fungi, plants use proline to inhibit their (fungal) non-self recognition reaction, reducing their (fungal) resistance to mycoviruses (Fig. 5C).”

Line 301-302 “This is beneficial to plants but detrimental to pathogenic fungi, therefore, plants defend themselves against phytopathogenic fungi. Delete this sentence.”

Line 416 use ‘Figures’ rather than ‘Figs’.

Figure 3H. Include the days post inoculation onto PDA for both the – and + proline treatments.

References need to start on a new line though this may be an error created through PDF generation as the Reference list appears twice.

REVIEWER COMMENTS

Reviewer #1 (Remarks to the Author):

The work submitted by Hai et al. focuses on a topic of considerable interest. Mycoviruses (fungal viruses) have gathered great attention as potential biocontrol agents of fungal diseases since the discovery of CHV1, a mycovirus that induces strong hypovirulence in its fungal host, *Cryphonectria parasitica*, and had spread naturally in the population of this pathogen in Europe. However, mycoviruses generally lack an extracellular route of transmission and they spread only by intracellular mechanisms. Fusion of vegetative hyphae is the major route of mycovirus horizontal transmission. Therefore, a major concern in the practical application of mycoviruses for the control of fungal diseases is that the complex vegetative compatibility/incompatibility systems developed by filamentous fungi to limit hyphal fusion will impair the spread of the mycoviruses introduced in the environment as biological control agents. Some previous data and observations suggested that environmental factors might promote mycovirus transmission even between incompatible strains. The results presented here provide compelling evidence that the plant environment does indeed weakens vegetative incompatibility barriers greatly enhancing mycovirus transmission. Furthermore, the mechanism by which the plant weakens vegetative incompatibility promoting mycovirus spread is elucidated. I consider these results of considerable interest as the success of mycoviruses as biocontrol agents is strongly dependent on their ability to spread in the population of the pathogen in natural conditions. What is more, these results can have practical application in the implementation of mycovirus-based strategies to control fungal diseases. The work is very complete as they have approached the topic from a number of angles, generating data that combined strongly support their findings.

Response: Thank you very much for your positive comments.

However, in spite of the merits of the research presented, the manuscript has many problems that need to be addressed. The text conveys a lack of attention to how things are expressed making it difficult to follow; one requires a high degree of previous knowledge on the topic to understand the relevance of the work, and even how many experiments were performed and therefore to properly evaluate the results. I have the strong feeling reading the current version of the manuscript that only readers very familiar with this type of research would be able to properly understand the work. Besides, the text conveys a lack of clarity of concepts and some of the information provided is inaccurate or wrong. I also have some concerns regarding some experiments and the quality of some images that I will refer to in the corresponding section.

Response: We greatly appreciate your valuable comments and constructive criticism of our research and writing, which helped us improve the quality and clarity of our manuscript, and also inspire us greatly.

We are very happy to edit the text further and supply more information, based on your helpful comments in this review. Thank you very much again.

Below I explain in more detail my concerns and provide comments and suggestion regarding each section of the manuscript.

INTRODUCTION

The authors could do a better job in the Introduction describing the role that fusion of vegetative hyphae has in the spread of mycoviruses. This would provide the necessary context to understand the relevance of their results that can be of great interest in positively re-assaying the potential of hypovirulence-inducing mycoviruses (HIVs) as biological control agents against fungal diseases. Information regarding the role of vegetative hyphae fusion as a major route of mycovirus horizontal transmission is dropped here and there but in a rather disorganized manner. As a matter of fact, some of the information included in the Introduction is not accurate. For example:

Lines 40-41: The fusion of vegetative compatible hyphae does not generally result in fusion of both nuclei generating a diploid nucleus, as stated, and to refer to this process as “mating” creates confusion. Sexual reproduction and the parasexual cycle of exchange of genetic material through the fusion of vegetative compatible hyphae are two distinct processes even though it can be some overlap in the molecular mechanisms that regulate them. Hyphal anastomosis is the major route of horizontal mycovirus transmission. Sexual reproduction, in those filamentous fungi with sexual cycle, doesn't seem to have a major role in the transmission of mycoviruses between strains. CHV1 for instances is not transmitted to sexual spores. They have to explain these things in a clear manner providing the necessary context for the research presented.

Response: Thank you very much for your comments which helped us expand our knowledge on the relationship between hyphae fusion and sexual reproduction. We have now deleted this inaccurate statement in the revised version of our manuscript. In the first paragraph of the introduction, we have now focused exclusively on vegetative or heterokaryon (in)compatibility, which is an important concept for our work, and not discussed sexual reproduction. Please see lines 40-42.

We hope that the revised version of our manuscript is now more clear and accessible to a non-specialised audience.

Lines 59-60: The authors state that mycovirus can be acquired via infection by viral particles and then transmitted by intracellular mechanisms. That's inaccurate, extracellular routes of transmission are not prevalent in mycoviruses, in most cases transmission only takes place through intracellular mechanisms like hyphal anastomosis (horizontal transmission) and sporulation (vertical transmission). Funnily enough, to support their statement that initial infection can take place through viral particles they cite a review article on CHV1, a capsidless mycovirus that doesn't therefore generate viral particles. The authors indeed admit that transmission through intracellular mechanisms is the key route of infection later on in the Discussion, when they provide the weakening of vegetative incompatibility mediated by proline generated by the plant as a putative explanation of the identification of the same mycovirus in different fungal species. If one thinks about it, this distinction between first acquired infection (that can be via viral particles) and then transmission (that takes place through hyphal anastomosis) makes no sense; when a mycovirus is transmitted by hyphal anastomosis the recipient strain is "acquiring infection". Thus, this statement conveys lack of clarity of ideas.

Response: Thank you for pointing this out. We have now deleted this statement in the revised version of our manuscript. We have also stated in the introduction that mycoviruses lack an extracellular phase in their replication cycle and are transmitted horizontally between fungal strains and vertically from parent to offspring, through respectively hyphal fusion or sporulation, please see lines 57-59.

The rest of this paragraph is also confusing; let's check for example this sentence "Despite the fact that VIC groups are extremely complex in filamentous fungi, identical mycoviruses have been frequently characterized in fungal isolates found in nature and belonging to different genera or species." First of all, what does it mean that a VIC group is extremely complex? I guess that what the authors mean is that filamentous fungi can have many different VIC groups, that is, it's the structure of VIC groups what is complex. To make this point clearer they could mention that within a fungal species, strains that harbor the same alleles in the genes that control this process are compatible to undergo hyphal anastomosis and are placed in the same VCG (vegetative compatibility group) whereas those that have different alleles are vegetatively incompatible and belongs to different VCGs. And then explain that the genetic control of hyphal anastomosis can be very complex resulting in very complex VCG structures with a high degree of vegetative incompatibility within the fungal species. In this way the concept of complexity regarding VIC can be properly conveyed, a concept that is quite relevant to the research presented here. There is another problem with this sentence, the fact that in a fungus the VCG structure is very complex (first part of the statement) affects how mycoviruses are spread within a fungal species not their transmission to other

fungal species (second part of the statement). Therefore, this sentence should be split in two, first stating that in some filamentous fungi mycovirus infection is very prevalent even though their VCG structure is very complex, suggesting that there are mechanisms in natural conditions that weaken vegetative incompatibility. Secondly, the identification of the same mycoviruses in different fungi further reinforces this idea suggesting that incompatibility can be weakened in nature to the point of sometimes allowing horizontal transmission even between different fungal species.

These points are not minor, they convey a lack of clarity of concepts that is very detrimental to the manuscript.

Response: We completely agree with your suggestions and have re-written these statements in the revised version of our manuscript. Please see lines 72-85.

The broad presence of mycoviruses in filamentous fungi with very complex VCG structures and the identification of the same mycoviruses as naturally infecting different fungal hosts, are observations that suggest that there are environmental factors that weaken vegetative incompatibility promoting mycovirus transmission. Besides, there are previous works suggesting that horizontal mycovirus transmission might be enhanced in the plant environment. The authors only cite them at the end of the Discussion; I consider those papers (Brusini et al. 2013 and Suzuki et al. 2021) should already be cited in the Introduction. The Introduction should clearly provide the context (and rationale) for the research presented here, that is: (1) The potential of mycoviruses that induced hypovirulence as biological control agents is strongly dependent on their ability to spread in the natural populations of the pathogen, (2) hyphal anastomosis of vegetative hyphae is the major route of mycovirus transmission, (3) the complex vegetative incompatibility systems developed by fungi is therefore a concern in the potential application of mycovirus-based strategies to disease control, (4) however, some observations and previous results suggested that mycovirus spread in natural conditions is more efficient than predicted by the high degree of vegetative incompatibility that might be present in fungal populations.

I'd also mention in the Introduction that in the implementation of mycovirus-based control strategies mycoviruses are introduced in the field through infected strains. The first time this appears in the manuscript is when they described their field experiment. This kind of key information helps readers that are not experts in this topic to most easily follow the work.

Response: Thank you for these important comments. In the revised version of our manuscript, we largely re-wrote and re-organised the introduction following your suggestions. We anticipate that the current version is clearer than the previous version. Please see lines 57-85.

RESULTS

Section “Plant-fungus interactions enhance mycovirus transmission”:

Line 95: After explaining that mycoviruses are readily transmitted to a compatible strain but not to incompatible strains on PDA, they refer to Fig. 1B. However, Fig. 1B illustrates transference of HIVs (hypovirulence-inducing mycoviruses) not on PDA but when strains are co-inoculated in planta, and it shows successful transmission between incompatible isolates. So, it is misleading to refer to Fig. 1B at this point. Also, the authors should mention in the text key information to understand the results on PDA which are illustrated in Fig. 1, that is that successful transmission on plates is illustrated by the recipient strain acquiring the abnormal phenotype of the virus donor strain.

Response: We re-wrote the statements about mycovirus transmission and hypovirulent phenotypes, and also adjusted the citation from “Fig. 1B” to “Fig. 1A”. Please see lines 119-120.

Line 99: Delete reference to Fig. 1C at this point; just keep references to Fig. 1B, 1C and 1D in line 101.

Response: Done. Please see lines 124-126.

Lines 103: They should direct the reader at this point to Supplementary Fig. 1; the doubts I have about how the experiment was performed were answered after checking Supplementary Fig. 1 later on when they refer to it in line 108.

Response: We completely agree with this suggestion and we have now added reference to Supplementary Fig. 1 in the revised manuscript. Please see lines 128-130.

Regarding Supplementary Fig. 1 I’d change a few things to make it clearer. To place Dual culture on the arrows is misleading. The co-inoculations are before the arrow and the arrows better illustrate the selection of the recipient strain on PDA + hyg after the co-inoculation of both strains. This sentence in the legend is very confusing: “The isolates were re-isolated from strain 1980m that was dual-cultured with Ep-1PN, and subcultured on PDA containing hygromycin...” What they do after co-cultivation is to specifically select the virus recipient strain by growing on PDA + hyg, as only the recipient strain harbor a hygR marker; this should be clear in the text.

I’d like to emphasize again that the authors tend to described the experiments performed in such a way that only researchers that do this type of assays themselves would be able to understand them.

Regarding specifically Supplementary Fig. 1, changes should be made to both the figure and the figure legend to better convey what they did, that is: (1) co-inoculate both the virus donor strain and an incompatible virus-free strain that had been labeled with a hygR marker, (2) re-isolate the fungus

from plant (or from PDA) specifically selecting the recipient strain by growing on PDA + hygR, (3) analyze presence of the mycoviruses in the isolated recipient strains. It seems they first grow on PDA + hyg to specifically select the recipient strain and then transfer to PDA to assess phenotype but I'm unclear about that.

Response: We sub-cultured and purified the recipient strain 1980m on PDA containing hygromycin after co-cultivation for three generations. The donor strain Ep-1PN without hygromycin-resistance gene cannot grow on PDA containing hygromycin, ensuring the strain 1980m labeled with hygromycin-resistance gene did not contaminate by strain Ep-1PN. Subsequently, we further assess the phenotypes of these isolates derived from recipient strain 1980m on the normal PDA.

We revised Supplementary Fig. 1 to make it more comprehensible and added more information in figure legend. Please see Supplementary Fig. 1.

I see different phenotypes on plate in Supplementary Fig. 1; it would help if the authors explain if single infections of each mycovirus induce alterations in growth and described them, as they did with the doubly infected strain.

Response: Based on the RT-PCR results, these strains exhibiting hypovirulent-associated phenotypes were usually co-infected by SsDRV and SsRVL.

It took me some time to understand why the RNA-seq results were split in two figures, Fig. 2 and Supplementary Fig. 2, until I noticed that in the results presented in Fig. 2 they combined the HIV-infected strain with an incompatible virus-free strain whereas in the results presented in Supplementary Fig. 2 both strains were uninfected (in this case they used the uninfected version of the HIV-infected strain). This is of interest, I assume they did that to determine if the HIVs themselves might have an effect on gene expression associated to VIC but nothing of that is explained or discussed in the text; the experimental set ups and the results obtained have to be clearly explained so they can be properly evaluated.

Response: We completely agree your comments, we designed two experiments: one involved co-cultivating 1980m with Ep-1PN infected by hypovirulence-associated mycoviruses (HAVs) as showed in Fig. 2, and the other involved interaction between 1980m and HAVs-free strain EP-1PNA367 as showed in Supplementary Fig. 2, which could exclude the influence of mycovirus infection. We supplied more information to clearly describe our results in the revised manuscript. Please see lines 139-141.

There are inconsistencies in the figures that might look minor but create confusion. For example, in Fig. 2 the panels say PDA whereas in Supplementary Fig. 2 it says Medium.

Response: In revision, we consistently use ‘PDA’ to indicate ‘media’ in the Supplementary Fig. 2, as shown in Fig. 2.

Also, I miss the inclusion of a compatible interaction as a control.

Response: We set a compatible interaction as a control, please see and Supplementary Fig. 2.

Section “Plant proline accumulation induced by fungal infection enhances mycovirus transmissibility”:

Line 131: After and post-inoculation don’t go well together as post means after. In fact, I’d change the sentence to just “....., was upregulated in both oilseed rape and *A. thaliana* Col-O infected with virulent virus-free strain Ep-1PNA367 (Supplementary Fig. 4A)” as the figure shows that the increase of expression starts at 7 hours but the time course is quite different in oilseed rape and *A. thaliana*.

Response: Thank you for your suggestion. We have now rephrased this sentence in the revised manuscript. Please see lines 165-166.

Lines 133-135: The hypothesis the authors outline here is not properly understood if they don’t start by saying that vegetative incompatibility has been associated to the induction of PCD that prevents the formation of viable hyphal bridges between both strains. There is another problem, the work illustrated in Fig. 3 does not test any hypothesis regarding inhibition of non-self recognition or PCD what it does is to confirm that the increased mycovirus transmission rates observed in planta are indeed mediated by proline produced by the plant. So, the hypothesis they are really testing here is that the enhanced mycovirus transmission observed in planta is mediated by proline produced by the plant and this is what they should state at this point.

Response: Thank you for your suggestion. We have added the statement: the fungal non-self recognition response results in programmed cell death (PCD) and accumulation of reactive oxygen species (ROS). We also mentioned that proline has been reported to reduce PCD reactions and inhibit the accumulation of ROS in fungi, followed by our rephrased hypothesis based on your suggestion. Please see lines 167-169.

Line 135-137: Very confusing sentence, I propose to change it to: “To test this hypothesis, we obtained an *A. thaliana* Col-0 mutant with a T-DNA insertion in *p5cs2* which was impaired in proline production during *S. sclerotiorum* infection (Fig. 3B)”

Response: We have changed this sentence following your suggestion in the revised manuscript, please see lines 171-173.

Line 140: I guess that what the authors mean by “but did not enhance resistance (Fig. 3E)” is that addition of proline in the inoculation of oilseed rape with the virulent strain did not induce any protection against infection. They should express this idea in a clearer way as it is important, it provides additional evidence that the effect of proline is associated to increased mycovirus transmission and not the induction of resistance in the plant as the effect of proline in decreasing disease severity is only observed when the HIV-infected strain is present.

Response: Thank you for the suggestion. We have elaborated on this statement following your suggestion, and we hope it is now clearer in the revised manuscript. Please see lines 176-179.

Line 141: Change “detected” to “tested for the presence of”. I take the opportunity here to note that there are other instances in the text where the terms used sound odd, and emphasize again that the entire manuscript should be thoroughly revised. The text requires too much revision for me to be able to suggest specific changes throughout.

Response: Changed. In addition, we have made extensive revisions for this manuscript with helping of native English speakers, aiming to make it more understandable. Please see line 180.

Lines 144-150: This paragraph is very confusing, it should be re-written. It is difficult to understand what is the point they’re trying to make. Looking at the figure what I see is confirmation in in vitro culture that proline enhances mycovirus transmission and this exactly what they should highlight in the text.

Response: Thank you for your valuable suggestions, and we completely agree your suggestions. In revision, we supplied the related content as “Subsequently, we determined the optimal concentration of proline in PDA for the promotion of mycovirus transmission in vitro”, and also re-organized this paragraph. Please see lines 184-192.

Lines 151-155: This paragraph is rather messy and therefore difficult to follow. I’d express things in a simpler manner along these lines:

“To test if the proline-mediated promotion of mycovirus transmissibility in *S. sclerotiorum* is a general phenomenon, we measured horizontal transmission rates of HIVs in five previously reported hypovirulent strains, SZ-150, SX276, Ep-1PN, WF-1, SCH733 and AH98 (Supplementary table 1 and Fig. 3I), to incompatible strains SG9 and 1980m. For that, each HIV-infected donor strain was co-cultured with virus-free recipient strains SG9 and 1980m, on PDA and on PDA plus 3.2 mM proline.”

Response: We have reworded this statement following your comments. Please see lines 193-198.

Section: “Proline promotes hyphal fusion, inhibits expression of vic-related genes, and reduces ROS production”

This is the section that can be considered to focus on testing the hypothesis stated in lines 133-135, so that information should be brought here. And like I said, it is important to mention that VIC has been associated to induction of PCD. In this section is where they really assay if proline has an effect on hyphal anastomosis.

Response: We reworded this paragraph; please see lines 206-211.

Regarding the title of this section, I find it somehow misleading. It seems to be claimed that any gene that plays a role in vegetative compatibility/incompatibility is a vic gene. This is not the case, vic genes are better described as those specifically associated to the process, the genetic determinants of the structure of VCGs (Vegetative Compatibility Groups) of a fungal species. In this section the authors perform a general study of changes in transcription patterns in response to proline. For example, G-protein signaling might play a key role in controlling hyphal fusion but that doesn't mean that the components of G-protein signaling pathways are vic genes. The same goes for the title of Fig. 4.

Response: We agreed your comments. We changed the title into “genes associated with VIC reaction”, and these genes include *Het* genes, G-protein genes, and ROS-related genes that could be involved in the process of hyphal fusion, please see line 202.

The inclusion of a compatible interaction in this study would've allowed to compare gene expression in the incompatible interaction in the presence of proline with gene expression in the compatible interaction.

Response: We compared the gene expression in the incompatible interaction in the presence of proline with gene expression in the compatible interaction. This information supplied in the revised manuscript. Please see lines 233-236 and Supplementary Fig. 7.

Line 165: They mention strain SCH733 but none of the results presented in either Fig. 4 or Supplementary Fig. 6 include this strain. The authors don't explain what the arrows point at in Supplementary Fig 6; I can see hyphal degradation in the upper panel of this figure but I don't know what the arrows point at in the lower panel.

Response: We supply information on SCH733 that was shown in Supplementary Fig. 5 and Supplementary Table 1, we also explain the arrows in Supplementary Fig 6. Please see legends of Supplementary Fig 6.

Lines 172-174: Fig. 4C is poor quality and to be honest I don't know what I'm supposed to look at in this figure. In that sense, the legend is little helpful, it states:

“PI staining and visualization using confocal microscopy for detection of cell fusion or cell death at the interface between 1980m and Ep-1PNA367G.”

What cell fusion are we seeing here? And again, they don't explain what the arrows mean. According to the M&M strain Ep-1PNA367G is the one labeled with GFP, so what cell fusion do we see under GFP fluorescence? In the presence of proline all the structures observed in the DIC image are also observed in the GFP image. All I can say in this figure is that there is no staining with PI in the presence of proline pointing to lack of cell death.

Response: We supplied more information into legends of Fig. 4C. When two incompatible stains (Ep-1PNA367G and 1980G) were contacted, and red signals will be strengthening at their interface, which suggested that cell death could happen, resulting in PI staining. However, no similar phenomenon was observed on the PDA containing proline, suggesting that the cell death did not happen and failed by PI staining. Please see lines 216-219 and legends of Fig. 4C.

Lines 174-176: Fig. 4D is really poor quality. If the authors want to be able to illustrate viable hyphal anastomosis between incompatible strains in the presence of proline with the two strains labelled with different fluorescent proteins, they really need to show better images. Also, the legend should be more informative. And like I said, it would add to the work to illustrate how the compatible interaction looks at the microscopic level and compared with the incompatible interaction plus proline.

Response: We have revised Fig. 4D to make it more understandable and added captions “the orientation of the white box indicates that the strain originally labeled with one fluorescent group showed a signal of another fluorescent group, suggesting the fusion of hyphae carrying different fluorescent labels.” Please see Fig. 4D and legends of Fig. 4D.

Compared to the incompatible interaction on PDA plus proline, the visibly necrotic zone is not significantly observed when two *S. sclerotiorum* strains are compatible contact on PDA (as shown in manuscript Figure 1A, Ep-1PN VS Ep-1PNA367), suggesting the PCD response of compatible interaction is significantly weaker than that of two incompatible interaction plus proline on PDA. We think there is no PCD reaction between compatible interaction on PDA based the results of RNA-seq about genes associated with VIC reaction (as shown in Supplementary Fig. 2 and Supplementary Fig. 7). Moreover, the mycovirus transmission efficiency should be 100% between two compatible strains Ep-1PN and Ep-1PNA367 on PDA (Xie et al. 2006, J Gen Virol. 87(Pt 1):241-249.), which is higher than that of two incompatible strains on PDA plus proline (as shown in Figure 3G). This

result further confirmed that proline only significantly weakens non-self recognition reaction between two compatible strains, but could not completely inhibition non-self recognition reaction. However, we could not observe the significantly different between compatible interaction and incompatible interaction plus proline on PDA at the microscopic level, as shown in Supplementary Fig. 6 (+proline) .

Line 178: It's not the gene expression profile of *S. sclerotiorum* but that of the interaction of two vegetative incompatible *S. sclerotiorum* strains. I emphasize again that more care should be taken to explain the work properly.

Response: We re-wrote the sentence accordingly for clarity in the revised manuscript and we hope it is clearer. Please see lines 222-224.

Lines 186-188: I assume the RNA-seq data in planta they compare here with the response to proline on PDA, is the same one presented in Fig 2 (first section of the results); they should make this clear. Besides, the references here to Supplementary Fig. 4B is confusing.

Response: Yes, these experiments were designed concurrently. We initially mentioned only the interaction with plants and PDA in Fig 2. Later, we introduced the proline treatment in Fig 4. Our goal was to identify genes related to fungal non-self recognition that were downregulated by both proline and plant interactions. We provided more information in Methods section "RNA extraction, RT-PCR, and RNA-seq"

We changed "Supplementary Fig. 4B" into "Fig. 2" in the revised manuscript. Please see line 233.

Line 193: Checking the Supplementary figure, I see that what the authors did was targeted deletion of the entire gene ORF, so it'd be more appropriated to say that they generated deletion mutants. And that all throughout the text. For example, in line 196 they should say "Ssvic2 deletion resulted in...."

Response: Revised throughout the manuscript. Please see lines 241-250.

Lines 200-203: Odd and confusing sentences. First, I think that the second sentence should go first, and then state that the highest increase in mycovirus transmission was observed in the mutant deleted for Ssvic3. Besides, the way the experiment was performed need to be clearly explained; I guess they co-cultivated the two strains and then specifically selected the recipient strains growing in the presence of neomycin, as the mutants harbor a neoR marker, and then they tested for the presence of the HIVs in the recipient strains. They say in the legend that N=24 for the co-cultures; does that

mean they analyze 24 samples per pair of donor and recipient strain? I'm guessing all this for what I know about this type of assays, for somebody with no experience in this type of experiments it would be even more difficult to understand.

Response: We re-organized these sentences following your comment, and more information of the related experiments was supplied. Please see lines 248-254.

'n=24' denotes 24 biological replicates for each co-cultivated experiment. Subsequently, the recipient strains were subcultured for purification on PDA containing neomycin, similar to the approach in Supplementary Fig. 1. Finally, we assessed for the presence of the HAVs in the recipient strains. We also added more information in figure legend (lines 731-734, Fig 1E)

There is another point to be made. The authors keep saying that strains were converted to hypovirulent when describing results where it is clear that they didn't do any virulence assay, and that's confusing. It would be clearer to say that strains exhibited the phenotypes associated to hypovirulence. They could also explain at some point early in the text that as infection by the HIVs under study induces clear alteration in the growth of the host fungus, these phenotypic alterations can be used as a visual marker of acquired infection.

Response: Thank you for your suggestion. We have reworded these sentences for clarity. Please see lines 254-255.

Line 203: There is no Fig. 4I, I think they mean Fig. 4J.

Response: Revised.

DISCUSSION

More attention should be paid to the structure of the Discussion, it looks somehow disorganized.

Line 217: I consider that the information provided in the second paragraph (lines 226-238) should be included at this point. This is the place to mention previous data suggesting that the plant could improve mycovirus transmission and how it is of considerable interest to study this further, to determine if it's a general phenomenon, and to elucidate by which mechanism the plant induces weakening of the incompatibility barriers.

Response: Thank you for your suggestion. We have re-organized this paragraph following your suggestion, please see lines 271-284.

Line 220: The statement "thus acting as a mechanism against fungal infection" could be a separate sentence explaining that as vegetative incompatibility is thought to have evolved to limit the spread of extrachromosomal genetic elements detrimental to the fungus such as mycoviruses and

senescence plasmids, the weakening of vegetative incompatibility reactions by the plant can act as a mechanism against fungal infection.

Response: Thank you for your suggestion. We have added these sentences. Please see lines 298-305.

Lines 220-225: The production of proline by the plant seems to be a response to infection. It would help if the authors provide more information on what is known about the role of plant proline during infection. It is true though that in the field assays, adding proline alone didn't have a protective effect against infection. It seems clear that the main effect on proline in decreasing disease severity was by enhancing HIVs transmission.

Response: We re-wrote this discussion part, including supplying more information on proline: Numerous studies have indicated that higher plants exhibit an elevation in proline content in response to various environmental stresses. Proline, functioning as a signaling molecule, modulates mitochondrial activities, influences cell proliferation or cell death, and triggers specific gene expression, which can be essential for the plant's recovery from stress. Please see lines 292-296.

The paragraph between lines 239 and 257 mixes information on different topics. The discussion about how zinc compounds were previously shown to weaken VIC but this is the first report of a plant-derived compound promoting hyphal anastomosis could be a separate paragraph. As for the information regarding identification of the mycoviruses in different fungal species, they can discuss this later, mentioning that it would be of interest to determine if proline can also promote hyphal anastomosis between different species.

Response: We deleted the discussion about zinc compounds. We re-wrote this section following your suggestion. Please see lines 285-305.

Lines 258-276: This part of the discussion is too speculative and sometimes difficult to follow. It is indeed of interest to discuss the case of CHV1 which is used for disease control in Europe but doesn't work well in North America, and how the results presented here provide an avenue to explore to try and improve the potential of CHV1 as a biocontrol agent. However, I suggest that this part of the discussion is re-written to increase clarity and avoid too much speculation. What the authors can safely claim is that even if the production of proline by the plant and its promotion of hypovirus transmission is a general mechanism, their data show that increasing the concentration of proline above the level produced by the plant in response to infection, by addition of exogenous proline, has a greater impact on hypovirus transmission. Therefore, they postulate that applying the appropriate amount of proline together with the CHV1-infected strains could significantly increase CHV1 transmission rate and it is something worth exploring.

Response: We accepted your suggestion and added more information following your suggestion in revision. Please see lines 358-372.

Lines 277-285: This paragraph is confusing. It must be a good idea to enumerate the different studies they performed that combined provide compelling evidence that proline alleviates non-self recognition in *S. sclerotium*, but I feel such information should be included earlier in the Discussion. On the other hand, if they mean to focus this paragraph on the genetic determinants of vegetative compatibility/incompatibility, they need to re-write it to better conveyed the point they want to make. The authors appear to argue that because the genetic regulation of vegetative compatibility/incompatibility is similar in different fungi, they have shown that proline is a general regulator of this process. They have no grounds to make such statement. At most they can argue that it would be interesting to investigate how proline affects the expression of genes that regulate hyphal anastomosis in fungi where the process is well studied and those genes have been identified. I guess they based their claim on the results presented in Supplementary Fig. 3 but those results only show that G-protein signaling is downregulated in fungi during infection. G-protein mediated signaling pathways play many roles in fungi, including roles in pathogenicity.

I should also mention that as in other parts of the manuscript, the information the authors provide about the genetic regulation of vegetative compatibility/incompatibility is not accurate. There are not a set of homologous genes governing VIC in filamentous fungi. The situation is more complex, genes regulating vegetative compatibility/incompatibility can be quite diverse in the different fungi. Even within a fungal species, the genes whose allelic differences determine vegetative compatibility/incompatibility are very diverse, have different molecular function and some of them have a HET domain and some don't. This is why identification of these genes is not straight forwards. A fungal species in which the genetic of VCGs is well characterized and illustrate the points I'm making is *C. parasitica* (see for example, Choi et al. 2012, Genetics 190: 113-127 and Zhang et al. 2014, Genetics 197: 701-714).

Response: Thank you very much for your suggestion. We moved this section to earlier place "the third paragraph" in the discussion.

In addition, we deleted the statement about G-protein in this part, and only discussed the reported genes (*Het* genes including *Ssvic*, *Ssvib1*, and *Sscwr*) that related to fungal cell fusion and CHV1 transmission. Please see lines 308-313.

Lines 286-295: I think this point is quite interesting but it is broader than stated by the authors and should be discussed in more detail. There is the long-standing question why mycoviruses that negatively affect fungal growth are still spread in nature, they are not lost even though they are

expected to be less fit ecologically than their virus-free counterparts. CHV1 for example has a clear negative impact on growth and sporulation and yet it had spread naturally in the populations of *C. parasitica* in Europe. Some mycoviruses don't have a good rate of vertical transmission to conidia and yet they're present in the natural populations of the fungus and are not lost over time. I think it is interesting to discuss this question in light of the results presented in this manuscript. The plant promoting horizontal transmission of these hypoviruses resulting in continuous new infections could make up for the lower ecological fitness of the strains harboring them.

Response: It is very interesting discussion, and we strongly agree with these views, and we combined those discussions in revision. Please see lines 335-357.

Lines 295-297: These statements make little sense and should be deleted or re-phrased so that the point the authors are trying to make here can be understood.

Response: Deleted.

Line 301: Delete reference to Fig. 5C, this is a general statement regarding the results provided here not something specific of Fig. 5C.

Response: Done.

MATERIALS AND METHODS

Line 311: By inoculating leaves with what? Looking at the pictures of oilseed rape I guess they use plugs of mycelia.

Response: Yes, we used plugs of mycelia. We added this information in revision. Please see lines 386-387.

Lines 317-318: The expression "genomic polymerase chain reaction (PCR)" sounds strange; I guess what they mean is PCR using plant genomic DNA as a template. Like I said before, the authors need to fix these problems that make the text look like informal talk.

Response: We carefully checked the whole manuscript, and re-wrote these similar sentences in revision. Please see line 393.

Line 323: Section "Mycovirus horizontal transmission assay." There is an abuse of information in parenthesis in this section.

Response: We re-wrote these sentences in the revised manuscript. Please see lines 399-400.

Line 330-331: I guess it'd be more accurate to say that "Reverse transcription (RT) followed by PCR with mycovirus-specific primers was used to assay infection in the recipient strains to quantify the

rate of mycovirus horizontal transmission (Supplementary table 2).” Also, the Supplementary Table 2 has to be fixed, terms like “Detection for” and “PCR for upstream fragment of” sound very odd.

Response: We fixed these statements in revision; please see line 405, and revised Supplementary Table. 2

Line 332: Section “Nucleic acid extraction, RT-PCR and RNA-seq analysis.” They say nucleic acid but they only talk about RNA, so they should say RNA extraction.

Response: Agreed, please see line 407.

Lines 33-334: The first sentence of this section is very confusing; it is clear that they want to include in just one sentence all the different biological materials they use for RT-PCR and RNA-seq but that doesn’t work. Either they split this information in several sentences so that it makes sense or just say that they extracted RNA from both plants and *S. clerotiorum* grown in vitro using a commerial..... I favor the first option.

Response: In the revised manuscript, we re-wrote these sentences, please see lines 408-410.

Line 368 (Section Microscopy): Which GFP and mCherry cassettes did they use for the expression of GFP and mCherry? To provide that information doesn’t take much room and makes the experiments easier to understand. Like I said, there is seemingly little care about how things are expressed throughout the text and in this section, there is a little example of that, the authors claim that the transformation resulted in fluorescent strains such and such; it would be more accurate to say that they named the resulting strains such and such.

Response: The information on cassettes of Green and mCherry fluorescent protein and their overexpression vectors were supplied in revision. Please see lines 448-450 and Supplementary Fig.11.

We have uploaded the NCBI database with more detailed informations of these vector sequences (Genbank accession number: PP209073 and PP209074).

Line 369: It should be specified “PEG-mediated protoplast transformation.”

Response: Changed. Please see line 451.

Line 374: Section “Generation of gene KO mutants.” Like I said before, they should talk about gene deletion mutants rather than KO mutants. Besides, whereas in most sections of the Materials and Methods they give little information, in this one they seem to provide too many details. This section could be shortened, for example from line 381 they could say “Equimolar amounts of the of the

purified NP and PT were used to transform protoplasts of *S. sclerotiorum* using a standard PEG-mediated protocol.” And then explain how transformants were selected. The last sentence “and were purified by single-protoplast isolation to obtain homozygous strains.” makes no sense to me; how single-protoplast isolation is going to be performed after you obtain your transformants?

Response: Thank you for your suggestion. We changed “KO mutants” into “gene deletion mutants” in the whole manuscript. Also, we completely agreed your comments, and shortened the section about protocol of transformation, and supplied more information on single-protoplast isolation. Please see lines 457 and 469-478.

Line 408: I guess that what the author mean is “earlier than the epidemic phase of stem rot” or “before the epidemic phase of stem rot”

Response: Changed into “earlier than the epidemic phase of stem rot”, please see line 496.

FIGURE LEGENDS

Figure legends in general do a poor job describing the result presented in the figures; they should be thoroughly revised so that the figures can be understood without having to check other sections.

Response: We have made extensive revisions to Figure legends; we hope this is now clearer.

Reviewer #2 (Remarks to the Author):

The manuscript titled “plants interfere with non-self recognition of fungi via proline accumulation to facilitate mycovirus transmission” detailed evidence supporting an observation that identical mycoviruses frequently found in fungal isolates in nature and belonging to different genera of species. Their data support that mycoviruses are less restricted by vegetative compatibility when the fungal hosts co-infecting plants than in vitro without plants. The authors presented multiple gene differential expression data, proline accumulation, mycovirus transmission rate, TEM, ROS, knockout mutants for genes potentially associated with proline’s sequestering activity on ROS upon anastomosis. Overall, this is a well-done and well-written manuscript.

Response: Thank you very much for your comments and suggestion.

One concern on the generality of this study described with their manuscript title, whether this claim applies to other fungi and mycovirus genotypes or can be demonstrated between *Sclerotinia sclerotiorum* and another fungal species in a cross-species manner. For example, whether SsDRV and SsRVL or the co-infecting SsEV1, SsHV1, SsNSRV1, SsMYRV4, and SsPV1, can be transmitted to another fungal species such as *Botrytis cinerea* in plants that are a common host to both pathogens. Some of these viruses may have been reported in a virome study of *B. cinerea*, and

it would be important to know if a transmission can be artificially induced by co-infecting them on plants or proline-amended PDA. There had been reports between the two fungi, although this reviewer does not have time to look into the overlapping members of viromes.

This reviewer recommends the authors to either include such data to support the claim or instead, specify that this work tested dsRNA mycovirus transmission between different *S. sclerotiorum* species that don't typically form anastomosis on usual PDA. In the later case, the title will need to be modified accordingly.

Response: We have strengthened the title. We changed “fungi” to “a plant pathogenic fungus”

As you mentioned, there are indeed some highly similar genomic sequences of mycoviruses between different fungal species and genera. For example: Two mitoviruses, Ophiostoma mitovirus 3a (OMV3a) and OMV3b co-infecting Ophiostoma novo-ulmi (Ophiostomataceae), naturally infect, respectively, two other distantly related fungal species *Sclerotinia homoeocarpa* and *Botrytis cinerea*. Moreover, a certain proportion of mycoviruses are shared in the mycoviromes of *B. cinerea* and *Sclerotinia sclerotium*, and mycoviruses are shared in mycoviromes of two phylogenetic distant fungal species *Leptosphaeria biglobosa* (Leptosphaeriaceae) and *B. cinerea* (Sclerotiniaceae) (Deng et al. 2022, ISME 16:2763-2774). Experimental evidence has shown that SsPV1 is transmitted interspecifically to other species (from *S. sclerotiorum* to *Sclerotinia nivalis* and *Sclerotinia minor*) (Xiao et al., J Virol. 2014, 88(17):10120-33). Therefore, current research supports the hypothesis that cross-species transmission of mycoviruses occurs frequently in nature, and the host range of mycovirus is wider than previously recognized.

In our group, we found, although, most hypovirulent-associated mycoviruses are challenging to infect other species through horizontal transmission, we have limited instances of successful interspecies horizontal transmission with the aid of plants. For example, a mycovirus infecting SCH733 (Hai et al. 2022, Phytopathology 112:2449-2461), *Sclerotinia sclerotiorum* mitovirus 36 (SsMV36), shows nearly 100% similarity with *Sclerotinia nivalis* mitovirus 1 infecting *S. nivalis* and *Botrytis cinerea* mitovirus 4 infecting *B. cinerea*, and we confirmed successful transmission of SsMV36 from *S. sclerotiorum* to *B. cinerea* when those two fungal strains were co-inoculated in their shared plants, but failed to cross-genus transmit on the proline-amended PDA (as shown in below Figure R1), suggesting that the potential plant-associated factors in plants are also beneficial to mycovirus cross-species or genus transmission, but the function of proline in mycovirus transmission between fungal species needs to be further explored.

Figure R1. Oilseed rape plants enhance SsMV6 transmission between *S. sclerotiorum* and *B. cinerea*.

(A) *S. sclerotiorum* strain SCH733 was dual-cultured with *B. cinerea* strain B05.10 labelled with hygromycin resistance gene on PDA (A), and on the living oilseed rape leaves (B). (C) and (D) Re-isolates picked up from the strain B05.10 side were subcultured on PDA containing hygromycin and assessed their colony morphology and mycoviruses infection on PDA. The transmission efficiency of SsMV36 is 0% on PDA and 18.2% on oilseed rape leaves.

Interestingly, we found that mycovirus FaMBV1 (*Fusarium asiaticum* mycobunyavirus 1, Huang et al., 2023, 97(1):e0138122) could be successfully transmitted between two *Fusarium* species (from *Fusarium asiaticum* to *Fusarium graminearum*) on proline-containing PDA (please see below), suggesting that proline enhancing mycovirus transmission should be common phenomenon in fungi. However, FaMBV1 does not exhibit stable replication in its new host *F. graminearum*. Therefore, mycoviruses need to adapt to new hosts, otherwise will be eliminated by RNAi or other unknown mechanisms. This result also revealed that cross-species transmission of mycoviruses occurs more frequently in nature than previously recognized, but some of them may have been subsequently eliminated due to a lack of adaptability in new fungal hosts. Therefore, whether proline can also promote hyphal anastomosis between different fungal species or genera remains to be determined, which is worth exploring. In the meantime, we have altered the title of our manuscript to render it less general.

Reviewer #3 (Remarks to the Author):

This original research describes a significant finding that deserves publication in this journal. The research is a breakthrough as it solves a puzzle as to where and how mycoviruses can transmit between fungal hosts that are themselves vegetatively incompatible. The research also provides novel avenues for application of their insights into field-based research. The research is sound methodologically with sufficient detail provided. The manuscript is well written and the figures are concisely presented. There are some minor edits suggested below.

Response: Thank you very much for your comments and suggestions, and also for your encouragement.

Introduction

Consider citing Arshed et al. 2023 (DOI: 10.1016/j.fgb.2023.103827) who described VIC locus genes Bcvic1 and Bcvic2 in *Botrytis cinerea*, that is more closely related to *Sclerotinia sclerotiorum* than the other fungi whose VIC genes are referenced in the Introduction

Response: The reference (Arshed et al. 2023) was added accordingly, please see line 44 and reference 12.

Line 243. Do the authors mean “transmitted” or “present” in the sentence “Interestingly, a number of mycoviruses are transmitted between species, genera, or even kingdoms”

Response: It should be “present”. We fixed these statements in revision, please see lines 331-332.

Line 269-270 “... also found that proline could inhibit expression of a serial (series?) Het genes related to fungal non-self recognition, and conformed (confirmed?) via knock-out experiments...”

Response: Revised. Please see lines 308-310.

Line 272-276 Therefore, although proline (is) induced by *C. parasitica* infection to enhance mycovirus transmission between incompatible strains, the concentration of proline in chestnut or other plants maybe (may be) too lower (low) to largely enhance mycoviruses (mycovirus’) transmission, and then (delete ‘then’) eventually fail to accomplish the desired control efficiency on chestnut blight in North America.”

Response: We deleted this statement in revision.

Line 277-278 “We provided compelling evidence to (the notion) support that proline alleviates non-self recognition reaction between two incompatible fungal strains in *S. sclerotiorum*.”

Response: Thank you for your suggestion. We modulated this statement, please see line 306.

Line 299-301 “In the process of combatting infection of pathogenic fungi, plants use proline to inhibit their (fungal) non-self recognition reaction, reducing their (fungal) resistance to mycoviruses (Fig. 5C).”

Response: Revised, please see lines 375-376.

Line 301-302 “This is beneficial to plants but detrimental to pathogenic fungi, therefore, plants defend themselves against phytopathogenic fungi. Delete this sentence.

Response: Deleted.

Line 416 use ‘Figures’ rather than ‘Figs’.

Response: Done, please see line 504.

Figure 3H. Include the days post inoculation onto PDA for both the – and + proline treatments.

Response: Done. Please see revised Figure 3H.

References need to start on a new line though this may be an error created through PDF generation as the Reference list appears twice.

Response: We deleted the duplicated references in revision.

REVIEWER COMMENTS

Reviewer #1 (Remarks to the Author):

As I stated when I revised the first version of the manuscript submitted by Dr. Xie et al., the data presented here can make a substantial contribution to understanding the mycovirus-fungus-plant interaction, and has the potential of impacting the practical application of mycoviruses inducing hypovirulence in the control of fungal diseases. I therefore considered the results presented here of great interest. However, in the whole I thought that the manuscript still required a lot of work to be in publishable form, and made comments highlighting the many concerns I had, and suggestions regarding how they could be addressed. On reviewing this second version of the manuscript, I can see that some problems have been properly addressed. For example, the manuscript conveys a better understanding of concepts relevant to the work than before and the Discussion is now well structured and focused on the important questions. However, I have to note that the manuscript still requires a fair amount of editing, and there are questions that need to be addressed, before it can be considered in publishable form.

I'll go again through each section of the manuscript but in less detail than when I revised the first versions of the manuscript as to not repeat points I already made.

ABSTRACT

Line 30-31: I think that the authors should tone down this statement. For example, changing "and will form the base" to "that have potential" or "that provides an avenue to explore"
They can safely state that the mechanism they uncovered as playing a role in defense against fungal diseases has potential in the developing of control strategies, or that is worth exploring its practical application in disease control strategies. However, to claim as fact that these results will lead to successful control of diseases in agricultural settings is an overstatement.

INTRODUCTION

This section is improved as compared with the previous version for example including a good description of the anastomosis of vegetative hyphae and its regulation. However, the text still needs much editing. I will provide a number of examples where editing is required but this is not meant as an exhaustive list, the entire text has to be carefully revised and edited to improve syntax, clarity and accuracy.

Line 41: delete "is blocked"

Line 46: This sentence sounds odd here; the fact that the genetic determinants of vegetative compatibility/incompatibility are termed vic or het does not illustrate potential for high level of complexity. I would remove the entire sentence, so the text should read:
"The genetic loci determining self/non-self recognition are traditionally termed het (heterokaryon) or vic (vegetative incompatibility); within a fungal species, strains that harbor the..."

Then they can introduce the idea that the number of vic loci is quite variable and there are fungal species with very complex VCG structures while others have a low number of different VCGs. The authors should also provide information specifically on *S. sclerotiorum*, the species their study is focused on.

Line 57: The sentence "Mycoviruses, the obligate parasite of fungi," sounds odd. I'd keep it rather as it was: "Mycoviruses are obligate parasites of fungi."

That means the rest of the sentence has to be edited too. I suggest: "Unlike viruses from plant and animals, mycoviruses lack an extracellular phase in their replication cycle, with rare exceptions."

Line 60-61: This sentence sounds odd; it is odd to talk about critical ecological roles and latent infections in the same sentence without any transition. The authors need to find a better way to convey this information. For example, they could start by stating that many mycovirus infections appear to be latent, that is they don't induce discernible phenotypic alterations in the host fungus. And then go into explaining that this is not always the case and include the information about hypo, hypervirulence and physiological alterations. After that they can conclude that mycoviruses therefore potentially play important ecological roles in nature. As a matter of fact, even mycoviruses that apparently induce cryptic infections might play roles that are not easily discernible with the type of experiments usually performed.

Line 63: Delete "great" (or considerable and leave great)

Line 64: Talking about virocontrol agents without having explained the meaning of virocontrol is confusing. The biological control of fungal diseases based on the use of HAVs have been termed virocontrol. They have to introduce the concept before referring to it as it is not a term so widespread.

Also the sentence "this premise is based on the notion.." (lines 64-65) is very strange. The text in lines 62-71 should be re-written for clarity bearing in mind the main points the authors want to convey:

1. Mycovirus conferring hypovirulence have potential as biological control agents against fungal diseases.
2. This is due to the fact that they are transmitted by fungal anastomosis and can therefore spread in the populations of the pathogen once introduced through infected strains
3. However, fungi have mechanisms that restrict fungal anastomosis that are expected to impaired the spread of HAVs and they are therefore a concern in their potential application in disease control

Lines 72-76: I think this information should be part of the previous paragraph rather than a separate paragraph. Also, it appears that the mostly unsuccessful attempts to control *C. parasitica* in North America through CHV1 is a complex issue with many contributing factors, so I'd tone down the statement made in lines 75-76. I propose something along these lines:

"It has been proposed that that the complex structure of *C. parasitica* VCGs in North America might be one of the factors contributing to the failure of CHV1-mediated control of chestnut blight."

Also, they should include a citation, for example a review article about CHV1-mediated disease control where this question is discussed.

Line 81: Delete "the" before highly.

Line 82: I'd delete "frequently"; there are not many reports of this phenomenon. Also, to say natural isolates is misleading, I guess what the authors mean is that highly similar mycoviruses have been identified as naturally infecting different species. They could just delete the word natural and leave the sentence as it is.

Line 85: Do the authors mean "genera"?

Line 98: Delete "the" before potential.

Also, like I said before, the authors should provide information about what is known regarding VIC in *S. sclerotiorum*. Many mycoviruses have been identified in this species and some of them have potential in virocontrol because they greatly impact virulence. However, the success of these mycoviruses in disease control is greatly dependent on their ability to spread in the populations of the pathogen and therefore on the self- non-self recognition systems of the host fungus that control hyphal anastomosis. Fungal species can greatly vary in the complexity of VCGs and therefore it would be expected in some of them the application of virocontrol pose more of a challenge. As far as I know,

S. sclerotiorum has a high degree of VIC. I think it is important to include this information as the current work addresses the question if species with strong population incompatibility are still amenable to mycovirus-based control strategies. Like I said, fungal species are very variable in the degree of VIC in their populations resulting in very different scenarios regarding the potential of mycoviruses to spread.

Line 103-106: This part is rather messy. I suggest changes along these lines (after explaining that *S. sclerotiorum* infection induces an increase in proline content in the plant):

"We provide evidence that proline weakens non-self recognition reactions in *S. sclerotiorum*.

Therefore, we uncovered the mechanism underpinning high levels...."

What I'm trying to say is that the authors should find the best way to convey the key information, that is:

1. HAVs transmission is significantly higher in planta in *S. sclerotiorum*
2. *S. sclerotiorum* Infection induces an increase in proline content in the plant
3. Using several different experimental approaches, it was shown that proline indeed weakens non-self recognition
4. Therefore, the mechanism behind the significant increase in HAVs transmission in planta was elucidated
5. And they showed this mechanism could be used to improve the potential of HAVs-based control of *S. sclerotiorum* and potentially other pathogens

In its current form I don't think the text clearly convey this key points.

RESULTS

As the rest of the manuscript, the Results section is improved but still requires thorough revision. I can make some comments and suggestions but cannot performed, as a reviewer, the work still needed to have the manuscript in publishable form. Therefore, I'd like to emphasize again that this is not meant as an exhaustive list of all the problems the text has.

Line 113-114: The expression "hypovirulent phenotypes" sounds odd. I would say:

"*S. sclerotiorum* strain Ep-1PN co-infected with *Sclerotinia sclerotiorum* debilitation-associated RNA virus (SsDRV) and *Sclerotinia sclerotiorum* RNA virus L (SsRV-L) exhibits several phenotypic traits associated to hypovirulence, including abnormal colony morphology, reduced growth rate and lower virulence towards oilseed rape."

Line 118-122: This part of the text is still messy:

In line 119 I would delete "visibly apparent (such as less growth rate)" The sentence should therefore read:

"... converting Ep-1PNA367 into a hypovirulent strain with the abnormal colony morphology and reduced growth rate (Fig. 1A)."

The next sentence is even more difficult to follow. I propose to edit it along these lines:

"However, on PDA the mycoviruses were rarely transmitted from Ep-1PN to vegetative incompatible strains 1980m, SG9, and SCH941A1 of *S. sclerotiorum*, as illustrated by the lack of phenotypic growth traits associated to hypovirulence in the recipient strains (Fig. 1A)."

Line 126: If the authors claim "these results revealed" this is not a hypothesis so they cannot say later "to test this hypothesis". I'd change "revealed" to "suggested"

Line 128-129: I understand what the authors meant but it is not properly expressed. I suggest:

"To test this hypothesis, we re-isolated strain 1980m, which is labelled with a hygromycin resistance marker, from plant lesions through three rounds of sub-culturing on PDA with hygromycin, and then tested for the presence of mycoviruses SsDRV and SsRVL by RT-PCR. The RT-PCR results confirmed

transmission of the mycoviruses to the incompatible strain, with the transmission rates... ”

Line 137-141: This part of the text is still messy. As in other parts of the manuscript, the problem seems to be that the authors try to condense a lot of information in a single statement; the result is this long almost unreadable sentence:

“To unravel the potential mechanism for enhanced mycovirus transmissibility in planta, the expression levels of genes related to self/non-self recognition reaction, including het genes and G protein genes, were compared between vegetative growth stage on PDA and infection stage on plants for that 1980m that was, respectively, co-cultured with HAVs-infected strain Ep-1PN (Fig. 2) and HAVs-free strain Ep-1PNA367 (Supplementary Fig. 2).”

Line 143: Change to “..as compared to their in vitro expression”

Lines 160-161: This sentence is a bit confusing. I propose to edit it along these lines:

“Proline levels increased 4-fold in response to infection of oilseed rape with HAV-infected hypovirulent strain Ep-1PN and 10-fold in response to infection with HAV-free virulent strain Ep-1PNA367 (Fig. 3A).”

Line 162-163: It appears that an increase in proline levels is a response to infection and the fact that the virulent HAV-free strain induces more severe infection would explain why proline levels are higher in this case.

Line 182-183: To include this sentence here as it creates confusion. I guess the authors mean that as the enhanced HAVs transmission is not observed in plants impaired in proline production, these results confirm that plant proline is promoting HAVs transmission. This is true but they have to find a way to better convey this point.

Line 184-185: I said that when I revised the first version of the manuscript, the main purpose of this part of the work is not to determine the optimal concentration of proline in vitro to promote mycovirus transmission, but to add more evidence that proline enhances mycovirus transmission. Therefore, they studied how proline added to the media would affect mycovirus transmission in vitro. So, this is not a good way to start this section. Of course, they have to try different concentrations of proline and work out the conditions of the experiment but this is not the purpose of the experiment.

Line 207: Delete “due to cellular apoptosis”. Also, “therefore” is not a good connector between both part of the sentence, and would split in two sentences:
“...of PCD. A typical necrotic zone was clearly observed....”

Lines 222-232: There are many examples of poor wording throughout the text, too many to be able to go over all of them. See for example sentences between lines 222-232. The authors don't need to explain to me what they mean, I know what they mean because I'm also familiar with RNA-seq, but it is very badly expressed. Like I said before, to split the information in shorter sentences rather than trying to include a lot of information in one sentence can help.

Line 234-236: It is good that they include this information (although again the wording is poor) but, how does the expression pattern of the incompatible interaction + proline compare with that of the compatible interaction? That is, to which extend addition of proline to the incompatible interaction induces a gene expression profile similar to a compatible interaction? I think this is the interesting point to highlight.

I'd would like to add that in their response to my comments on the first version of the manuscript, the authors explain that their results show that proline weakens incompatible reactions but still there are

differences between the compatible reaction and the incompatible reaction + proline. I consider that the information they provide in the response to my comment is relevant and should be included in the manuscript. I'm referring to this information:

Compared to the incompatible interaction on PDA plus proline, the visibly necrotic zone is not significantly observed when two *S. sclerotiorum* strains are compatible contact on PDA (as shown in manuscript Figure 1A, Ep-1PN VS Ep-1PNA367), suggesting the PCD response of compatible interaction is significantly weaker than that of two incompatible interaction plus proline on PDA. We think there is no PCD reaction between compatible interaction on PDA based the results of RNA-seq about genes associated with VIC reaction (as shown in Supplementary Fig. 2 and Supplementary Fig. 7). Moreover, the mycovirus transmission efficiency should be 100% between two compatible strains Ep-1PN and Ep-1PNA367 on PDA (Xie et al. 2006, J Gen Virol. 87(Pt 1):241-249.), which is higher than that of two incompatible strains on PDA plus proline (as shown in Figure 3G). This result further confirmed that proline only significantly weakens non-self recognition reaction between two compatible strains, but could not completely inhibition non-self recognition reaction. However, we could not observe the significantly different between compatible interaction and incompatible interaction plus proline on PDA at the microscopic level, as shown in Supplementary Fig. 6 (+proline) .

Lines 240-244: Again, poor wording. I propose editing to the text along these lines:

"To examine whether any of the genes whose expression is down-regulated by proline in vitro and/or in planta play a role in mycovirus horizontal transmission, we generated deletion mutants for a subgroup of them in the genetic background of strain 1980. This deletion mutant analysis included genes *Ssvc1* to *Ssvc6*, and *Ssvib1*, *Sscwr1*, *Sscwr2*, and *Sscwr3*, which encode proteins containing HET/NT80 or LMPO conserved domains (Supplementary Fig. 9 and 10)."

Line 257: This heading illustrates the problems I see with the text as I don't think it properly captures the main idea of this section, that is, that although plants increase proline production in response to pathogens and that in turn enhances HAVs transmission, addition of exogenous proline further enhances the impact of HAVs in lowering disease severity and improving crop yield.

DISCUSSION

I consider that the Discussion is clearly improved, it is well structured and the points discussed are relevant. However, the text, like in the rest of the manuscript, requires a fair amount of editing. I will make a number of comments and give some examples but the level edition the text requires goes beyond what a reviewer can do.

Line 281-281: I would tone down these statements or re-phrase them differently to more clearly convey the information. As I see it the main points to convey here are:

1. Plants appear to weaken vegetative incompatibility in fungi which is the main barrier to mycovirus horizontal transmission.
2. This might have an important impact in the ecology of mycoviruses, and it is also important from a practical point of view regarding the application of mycoviruses in biological control
3. Therefore, it is of considerable interest to elucidate the mechanisms underline the weakening of incompatibility barriers by the plant.

Lines 289-292: I think this part is very confusing. A plant factor that promote transmission in planta? Then they compare nature with in vitro? Not plant with in vitro that is what can be compared by the experiments performed but results of in vitro transmission assays with observations in nature. I would make the statemen simpler and more to the point:

"This data, to the best of our knowledge, provides the first example of a plant

associated factor shown to promote mycovirus horizontal transmission, and points to a potential novel mechanism of plant defense against phytopathogenic fungi.”

Lines 313-315: I don't understand this sentence; which genes are the authors talking about?

Lines 316-334: This paragraph is very confusing; it seems the authors are mixing questions that are quite different and therefore it is difficult to understand exactly which points are they trying to highlight here. The entire paragraph should be re-written. I guess the paragraph is about factors known, or predicted, to affect VIC reactions but the information is all mixed.

Line 335: HAVs are by definition that, mycoviruses that induce hypovirulence (decreased virulence), so they do decrease virulence. I guess that what the author mean is that “HAVs have potential as biological control agents but questions regarding their ability to spread in the natural populations of the pathogen and ecological fitness under field conditions have to be addressed to expand their application in disease management strategies.”

Line 343-344: Very convoluted sentence. I suggest:
“Recent research has shown that HAVs can induce a change in the host fungus life style.”

Line 360-363: This part of the text is confusing. So, is there a paper showing that *C. parasitica* infection increases proline production in chestnut? They cite a paper (Kovács, G. E. et al.) but express it an odd way “*C. parasitica* could induce”. Assuming that plants infected with *C. parasitica* have increases levels of proline, I suggest to edit the text along these lines:

“Previous works show that *C. parasitica* infection increases proline production in chestnuts⁶⁸ and also that CHV1 transmission rates are higher in planta than in vitro^{62,69-72}. We hypothesize that the enhanced transmission of CHV1 in chestnuts is mediated by plant-proline, similarly to what we observed with the transmissions of HAVs in *S. sclerotiorum*.”

The rest of this paragraph (lines 363-372) includes very relevant information but poorly expressed; the entire paragraph should be re-written. An example of poor wording can be found in this long sentence:

“In our research, we experimentally confirmed that the concentration of proline induced by *S. sclerotiorum* infection is lower approximately 10-fold lower as compared to exogenously added proline in PDA and field experiments, and the transmission rate of mycoviruses is significantly higher on PDA with 3.2 mM proline than that observed on plant leaves without exogenously added proline.”

Like I said before, in addition to pay more attention to syntax, I think the authors should avoid trying to condense a lot of information in just one sentence.

Also, as they started the paragraph talking about CHV1, I think they should mention that their results provide an avenue to explore in trying to improve CHV1-mediated control of chestnut blight in North America, the addition of exogenous proline.

MATERIAL AND METHODS

Although the clarity of some parts of the M&M have improved with the new information, the text still requires work. I'm not going to go again through each section, I'll give only one example to highlight what I mean; the text in lines 408-413 were we find one of those very messy and long sentences in which the authors try to include a lot of different information:

“All *S. sclerotiorum* strains were cultured alone or together with an (in)compatibility strain on a cellophane membrane overlaying PDA with or without proline, or inoculated alone or co-inoculated on oilseed rape plants or *Arabidopsis*, and the mycelium was respectively collected and then total RNA was purified using a commercial RNA extraction kit (Takara, Code No.: 9108) for detection the expression of VIC-related genes with two methods, including the quantitative RT-PCR (RT413 qPCR) and RNA-seq.”

In addition to being messy, the sentence includes odd wording like "for detection the expression of VIC..."

Later all they say again "to detect the expression profile" (line 145); expression profiles are not detected, they can say for example to analyze or to study.

All these problems have to be fixed before the manuscript is in publishable form.

My final comment is regarding Figure 4D, I'm still concerned with the quality of these images and have my doubts they illustrate hyphal fusion as the authors claim. They authors have not changes the images so I guess they couldn't obtain others during the experiments that better illustrate their point. All authors have done is to add arrows pointing at the hyphae of the two strains and they added this comment:

"the orientation of the white box indicates that the strain originally labeled with one fluorescent group showed a signal of another fluorescent group, suggesting the fusion of hyphae carrying different fluorescent labels"

This sentence doesn't make sense to me. And all I see is what appears to be partial overlap of hyphae from different strains.

Reviewer #2 (Remarks to the Author):

My previous comments have been addressed.

Response points by points

Reviewer #1 (Remarks to the Author):

As I stated when I revised the first version of the manuscript submitted by Dr. Xie et al., the data presented here can make a substantial contribution to understanding the mycovirus-fungus-plant interaction, and has the potential of impacting the practical application of mycoviruses inducing hypovirulence in the control of fungal diseases. I therefore considered the results presented here of great interest. However, in the whole I thought that the manuscript still required a lot of work to be in publishable form, and made comments highlighting the many concerns I had, and suggestions regarding how they could be addressed. On reviewing this second version of the manuscript, I can see that some problems have been properly addressed. For example, the manuscript conveys a better understanding of concepts relevant to the work than before and the Discussion is now well structured and focused on the important questions. However, I have to note that the manuscript still requires a fair amount of editing, and there are questions that need to be addressed, before it can be considered in publishable form.

I'll go again through each section of the manuscript but in less detail than when I revised the first versions of the manuscript as to not repeat points I already made.

Response: Thank you very much for your detailed work on our manuscript. We are happy to further revise our manuscript following your suggestions.

ABSTRACT

Line 30-31: I think that the authors should tone down this statement. For example, changing “and will form the base” to “that have potential” or “that provides an avenue to explore”. They can safely state that the mechanism they uncovered as playing a role in defense against fungal diseases has potential in the developing of control strategies, or that is worth exploring its practical application in disease control strategies. However, to claim as fact that these results will lead to successful control of diseases in agricultural settings is an overstatement.

Response: we revised our statement. Please see lines 31-35.

INTRODUCTION

This section is improved as compared with the previous version for example including a good description of the anastomosis of vegetative hyphae and its regulation. However, the text still needs much editing. I will provide a number of examples where editing is required but this is not meant as an exhaustive list, the entire text has to be carefully revised and edited to improve syntax, clarity and accuracy.

Line 41: delete “is blocked”

Response: Done.

Line 46: This sentence sounds odd here; the fact that the genetic determinants of vegetative compatibility/incompatibility are termed vic or het does not illustrate potential for high level of complexity. I would remove the entire sentence, so the text should read:

“The genetic loci determining self/non-self recognition are traditionally termed het (heterokaryon) or vic (vegetative incompatibility); within a fungal species, strains that harbor the....”

Response: Done, please see line 49-53.

Then they can introduce the idea that the number of vic loci is quite variable and there are fungal species with very complex VCG structures while others have a low number of different VCGs. The authors should also provide information specifically on *S. sclerotiorum*, the species their study is focused on.

Response: Done. More information was supplied in revision. Please see lines 106-110.

Line 57: The sentence “Mycoviruses, the obligate parasite of fungi,” sounds odd. I’d keep it rather as it was: “Mycoviruses are obligate parasites of fungi.”

That means the rest of the sentence has to be edited too. I suggest: “Unlike viruses from plant and animals, mycoviruses lack an extracellular phase in their replication cycle, with rare exceptions.”

Response: Done. Please see lines 60-61.

Line 60-61: This sentence sounds odd; it is odd to talk about critical ecological roles and latent infections in the same sentence without any transition. The authors need to find a better way to convey this information. For example, they could start by stating that many mycovirus infections appear to be latent, that is they don’t induce discernible phenotypic alterations in the host fungus. And then go into explaining that this is not always the case and include the information about hypo, hypervirulence and physiological alterations. After that they can conclude that mycoviruses therefore potentially play important ecological roles in nature. As a matter of fact, even mycoviruses that apparently induce cryptic infections might play roles that are not easily discernible with the type of experiments usually performed.

Response: we re-wrote those statements in revision. Please see lines 63-69.

Line 63: Delete “great” (or considerable and leave great)

Response: Done.

Line 64: Talking about virocontrol agents without having explained the meaning of virocontrol is confusing. The biological control of fungal diseases based on the use of HAVs have been termed virocontrol. They have to introduce the concept before referring to it as it is not a term so widespread.

Response: Changed into “biological control” in the whole manuscript, please see 69-71.

Also the sentence “this premise is based on the notion..” (lines 64-65) is very strange. The text in lines 62-71 should be re-written for clarity bearing in mind the main points the authors want to convey: 1. Mycovirus conferring hypovirulence have potential as biological control agents against fungal

diseases. 2. This is due to the fact that they are transmitted by fungal anastomosis and can therefore spread in the populations of the pathogen once introduced through infected strains. 3. However, fungi have mechanisms that restrict fungal anastomosis that are expected to impair the spread of HAVs and they are therefore a concern in their potential application in disease control

Response: We re-wrote this section. Please see lines 69-82.

Lines 72-76: I think this information should be part of the previous paragraph rather than a separate paragraph. Also, it appears that the mostly unsuccessful attempts to control *C. parasitica* in North America through CHV1 is a complex issue with many contributing factors, so I'd tone down the statement made in lines 75-76. I propose something along these lines: "It has been proposed that the complex structure of *C. parasitica* VCGs in North America might be one of the factors contributing to the failure of CHV1-mediated control of chestnut blight."

Response: Completely agree, please see 71-76.

Also, they should include a citation, for example a review article about CHV1-mediated disease control where this question is discussed.

Response: Done. We cited "Milgroom and Cortesi, 2004". Please see line 73.

Line 81: Delete "the" before highly.

Response: Deleted.

Line 82: I'd delete "frequently"; there are not many reports of this phenomenon. Also, to say natural isolates is misleading, I guess what the authors mean is that highly similar mycoviruses have been identified as naturally infecting different species. They could just delete the word natural and leave the sentence as it is.

Response: Deleted, and re-wrote this sentence. Please see lines 85-87.

Line 85: Do the authors mean "genera"?

Response: Yes, we changed "general" into "genera", please see 89.

Line 98: Delete "the" before potential.

Response: Deleted.

Also, like I said before, the authors should provide information about what is known regarding VIC in *S. sclerotiorum*. Many mycoviruses have been identified in this species and some of them have potential in virocontrol because they greatly impact virulence. However, the success of these mycoviruses in disease control is greatly dependent on their ability to spread in the populations of the pathogen and therefore on the self- non-self recognition systems of the host fungus that control hyphal anastomosis. Fungal species can greatly vary in the complexity of VCGs and therefore it would be expected in some of them the application of virocontrol pose more of a challenge. As far as I know, *S. sclerotiorum* has a high degree of VIC. I think it is important to include this information as the current work addresses the question if species with strong population incompatibility are still

amenable to mycovirus-based control strategies. Like I said, fungal species are very variable in the degree of VIC in their populations resulting in very different scenarios regarding the potential of mycoviruses to spread.

Response: We supplied more information. Please see lines 106-114.

Line 103-106: This part is rather messy. I suggest changes along these lines (after explaining that *S. sclerotiorum* infection induces an increase in proline content in the plant): “We provide evidence that proline weakens non-self recognition reactions in *S. sclerotiorum*. Therefore, we uncovered the mechanism underpinning high levels.....”

Response: we followed your comments and revised the manuscript. Please see lines 117-121.

What I’m trying to say is that the authors should find the best way to convey the key information, that is:

1. HAVs transmission is significantly higher in planta in *S. sclerotiorum*
2. *S. sclerotiorum* Infection induces an increase in proline content in the plant
3. Using several different experimental approaches, it was shown that proline indeed weakens non-self recognition
4. Therefore, the mechanism behind the significant increase in HAVs transmission in planta was elucidated
5. And they showed this mechanism could be used to improve the potential of HAVs-based control of *S. sclerotiorum* and potentially other pathogens

In its current form I don’t think the text clearly convey this key points.

Response: Thank you very much for this clear information to understand our research. We try our best to revise the whole manuscript following those nice suggestions.

RESULTS

As the rest of the manuscript, the Results section is improved but still requires thorough revision. I can make some comments and suggestions but cannot performed, as a reviewer, the work still needed to have the manuscript in publishable form. Therefore, I’d like to emphasize again that this is not meant as an exhaustive list of all the problems the text has.

Line 113-114: The expression “hypovirulent phenotypes” sounds odd. I would say: “*S. sclerotiorum* strain Ep-1PN co-infected with *Sclerotinia sclerotiorum* debilitation-associated RNA virus (SsDRV) and *Sclerotinia sclerotiorum* RNA virus L (SsRV-L) exhibits several phenotypic traits associated to hypovirulence, including abnormal colony morphology, reduced growth rate and lower virulence towards oilseed rape.”

Response: Done. Please see lines 128-131.

Line 118-122: This part of the text is still messy: In line 119 I would delete “visibly apparent (such as less growth rate)” The sentence should therefore read: “.... converting Ep-1PNA367 into a

hypovirulent strain with the abnormal colony morphology and reduced growth rate (Fig. 1A).”

Response: Done. Please see lines 136-137.

The next sentence is even more difficult to follow. I propose to edit it along these lines: “However, on PDA the mycoviruses were rarely transmitted from Ep-1PN to vegetative incompatible strains 1980m, SG9, and SCH941A1 of *S. sclerotiorum*, as illustrated by the lack of phenotypic growth traits associated to hypovirulence in the recipient strains (Fig. 1A).”

Response: Done. Please see lines 137-139.

Line 126: If the authors claim “these results revealed” this is not a hypothesis so they cannot say later “to test this hypothesis”. I’d change “revealed” to “suggested”.

Response: Done. Please see lines 143.

Line 128-129: I understand what the authors meant but it is not properly expressed. I suggest: “To test this hypothesis, we re-isolated strain 1980m, which is labelled with a hygromycin resistance marker, from plant lesions through three rounds of sub-culturing on PDA with hygromycin, and then tested for the presence of mycoviruses SsDRV and SsRVL by RT-PCR. The RT-PCR results confirmed transmission of the mycoviruses to the incompatible strain, with the transmission rates....”

Response: Done. Please see lines 145-150.

Line 137-141: This part of the text is still messy. As in other parts of the manuscript, the problem seems to be that the authors try to condense a lot of information in a single statement; the result is this long almost unreadable sentence: “To unravel the potential mechanism for enhanced mycovirus transmissibility in planta, the expression levels of genes related to self/non-self recognition reaction, including het genes and G protein genes, were compared between vegetative growth stage on PDA and infection stage on plants for that 1980m that was, respectively, co-cultured with HAVs-infected strain Ep-1PN (Fig. 2) and HAVs-free strain Ep-1PNA367 (Supplementary Fig. 2).”

Response: we re-wrote this part and also tried to clearly statement in the whole manuscript, please see 155.

Line 143: Change to “..as compared to their in vitro expression”

Response: Done. Please see lines 172-173.

Lines 160-161: This sentence is a bit confusing. I propose to edit it along these lines:

“Proline levels increased 4-fold in response to infection of oilseed rape with HAV-infected hypovirulent strain Ep-1PN and 10-fold in response to infection with HAV-free virulent strain Ep-1PNA367 (Fig. 3A).”

Response: Done. Please see lines 177-178.

Line 162-163: It appears that an increase in proline levels is a response to infection and the fact that the virulent HAV-free strain induces more severe infection would explain why proline levels are higher in this case.

Response: Done. Please see lines 178-181.

Line 182-183: To include this sentence here as it creates confusion. I guess the authors mean that as the enhanced HAVs transmission is not observed in plants impaired in proline production, these results confirm that plant proline is promoting HAVs transmission. This is true but they have to find a way to better convey this point.

Response: Done, please see 199-200.

Line 184-185: I said that when I revised the first version of the manuscript, the main purpose of this part of the work is not to determine the optimal concentration of proline in vitro to promote mycovirus transmission, but to add more evidence that proline enhances mycovirus transmission. Therefore, they studied how proline added to the media would affect mycovirus transmission in vitro. So, this is not a good way to start this section. Of course, they have to try different concentrations of proline and work out the conditions of the experiment but this is not the purpose of the experiment.

Response: Thank you very much for your suggestion, we modified the statement in revision. Please see lines 201-210.

Line 207: Delete “due to cellular apoptosis”. Also, “therefore” is not a good connector between both part of the sentence, and would split in two sentences: “...of PCD. A typical necrotic zone was clearly observed.....”

Response: Completely accepted, please see lines 223-227.

Lines 222-232: There are many examples of poor wording throughout the text, too many to be able to go over all of them. See for example sentences between lines 222-232. The authors don't need to explain to me what they mean, I know what they mean because I'm also familiar with RNA-seq, but it is very badly expressed. Like I said before, to split the information in shorter sentences rather than trying to include a lot of information in one sentence can help.

Response: we re-wrote this section in the revision, please see lines 241-249.

Line 234-236: It is good that they include this information (although again the wording is poor) but, how does the expression pattern of the incompatible interaction + proline compare with that of the compatible interaction? That is, to which extent addition of proline to the incompatible interaction induces a gene expression profile similar to a compatible interaction? I think this is the interesting point to highlight. I'd would like to add that in their response to my comments on the first version of the manuscript, the authors explain that their results show that proline weakens incompatible reactions but still there are differences between the compatible reaction and the incompatible reaction + proline. I consider that the information they provide in the response to my comment is relevant and should be included in the manuscript. I'm referring to this information:

Compared to the incompatible interaction on PDA plus proline, the visibly necrotic zone is not significantly observed when two *S. sclerotiorum* strains are compatible contact on PDA (as shown in

manuscript Figure 1A, Ep-1PN VS Ep-1PNA367), suggesting the PCD response of compatible interaction is significantly weaker than that of two incompatible interaction plus proline on PDA. We think there is no PCD reaction between compatible interaction on PDA based the results of RNA-seq about genes associated with VIC reaction (as shown in Supplementary Fig. 2 and Supplementary Fig. 7). Moreover, the mycovirus transmission efficiency should be 100% between two compatible strains Ep-1PN and Ep-1PNA367 on PDA (Xie et al. 2006, J Gen Virol. 87(Pt 1):241-249.), which is higher than that of two incompatible strains on PDA plus proline (as shown in Figure 3G). This result further confirmed that proline only significantly weakens non-self recognition reaction between two compatible strains, but could not completely inhibition non-self recognition reaction. However, we could not observe the significantly different between compatible interaction and incompatible interaction plus proline on PDA at the microscopic level, as shown in Supplementary Fig. 6 (+proline).

Response: Thank you very much for your suggestions. We integrated the above information into the main text. Please see lines 226-233 and lines 251-254.

Lines 240-244: Again, poor wording. I propose editing to the text along these lines: “To examine whether any of the genes whose expression is down-regulated by proline in vitro and/or in planta play a role in mycovirus horizontal transmission, we generated deletion mutants for a subgroup of them in the genetic background of strain 1980. This deletion mutant analysis included genes Ssvic1 to Ssvic6, and Ssvib1, Sscwr1, Sscwr2, and Sscwr3, which encode proteins containing HET/NT80 or LMPO conserved domains (Supplementary Fig. 9 and 10).”

Response: Completely accepted, please see lines 260-264.

Line 257: This heading illustrates the problems I see with the text as I don't think it properly captures the main idea of this section, that is, that although plants increase proline production in response to pathogens and that in turn enhances HAVs transmission, addition of exogenous proline further enhances the impact of HAVs in lowering disease severity and improving crop yield.

Response: we slightly modified the heading illustrates following your suggestion, please see line 276.

DISCUSSION

I consider that the Discussion is clearly improved, it is well structured and the points discussed are relevant. However, the text, like in the rest of the manuscript, requires a fair amount of editing. I will make a number of comments and give some examples but the level edition the text requires goes beyond what a reviewer can do.

Line 281-281: I would tone down these statements or re-phrase them differently to more clearly convey the information. As I see it the main points to convey here are:

1. Plants appear to weaken vegetative incompatibility in fungi which is the main barrier to mycovirus horizontal transmission.

2. This might have an important impact in the ecology of mycoviruses, and it is also important from a practical point of view regarding the application of mycoviruses in biological control.

3. Therefore, it is of considerable interest to elucidate the mechanisms underline the weakening of incompatibility barriers by the plant.

Response: We followed your nice suggestions and revised the statements in our manuscript. Please see lines 292-302.

Lines 289-292: I think this part is very confusing. A plant factor that promote transmission in planta? Then they compare nature with in vitro? Not plant with in vitro that is what can be compared by the experiments performed but results of in vitro transmission assays with observations in nature. I would make the statemen simpler and more to the point: “This data, to the best of our knowledge, provides the first example of a plant associated factor shown to promote mycovirus horizontal transmission, and points to a potential novel mechanism of plant defense against phytopathogenic fungi.”

Response: Changed following your suggestion. Please see lines 307-309.

Lines 313-315: I don’t understand this sentence; which genes are the authors talking about?

Response: We revised this sentence. Please see lines 328-330.

Lines 316-334: This paragraph is very confusing; it seems the authors are mixing questions that are quite different and therefore it is difficult to understand exactly which points are they trying to highlight here. The entire paragraph should be re-written. I guess the paragraph is about factors known, or predicted, to affect VIC reactions but the information is all mixed.

Response: We almost re-wrote this paragraph, and deleted some information that is not closely related to the current research Please see lines 333-344.

Line 335: HAVs are by definition that, mycoviruses that induce hypovirulence (decreased virulence), so they do decrease virulence. I guess that what the author mean is that “HAVs have potential as biological control agents but questions regarding their ability to spread in the natural populations of the pathogen and ecological fitness under field conditions have to be addressed to expand their application in disease management strategies.”

Response: Completely agree your suggestion and revised. Please see lines 345-347.

Line 343-344: Very convoluted sentence. I suggest: “Recent research has shown that HAVs can induce a change in the host fungus life style.”

Response: Done, please see lines 353.

Line 360-363: This part of the text is confusing. So, is there a paper showing that *C. parasitica* infection increases proline production in chestnut? They cite a paper (Kovács, G. E. et al.) but express it an odd way “*C. parastica* could induce”. Assuming that plants infected with *C. parasitica* have increases levels of proline, I suggest to edit the test along these lines: “Previous works show that *C. parasitica* infection increases proline production in chestnuts 68 and also that CHV1 transmission

rates are higher in planta than in vitro 62,69-72. We hypothesize that the enhanced transmission of CHV1 in chestnuts is mediated by plant-proline, similarly to what we observed with the transmissions of HAVs in *S. sclerotiorum*.”

Response: Revised. Please see lines 369-372.

The rest of this paragraph (lines 363-372) includes very relevant information but poorly expressed; the entire paragraph should be re-written. An example of poor wording can be found in this long sentence: “In our research, we experimentally confirmed that the concentration of proline induced by *S. sclerotiorum* infection is lower approximately 10-fold lower as compared to exogenously added proline in PDA and field experiments, and the transmission rate of mycoviruses is significantly higher on PDA with 3.2 mM proline than that observed on plant leaves without exogenously added proline.”

Response: We re-wrote this paragraph, please see lines 372-379.

Like I said before, in addition to pay more attention to syntax, I think the authors should avoid trying to condense a lot of information in just one sentence.

Response: We tried our best to avoid syntax errors, and expressed our results with shorten sentence.

Also, as they started the paragraph talking about CHV1, I think they should mention that their results provide an avenue to explore in trying to improve CHV1-mediated control of chestnut blight in North America, the addition of exogenous proline.

Response: Thank you very much, we mentioned this information in revision, please see lines 383-386.

MATERIAL AND METHODS

Although the clarity of some parts of the M&M have improved with the new information, the text still requires work. I’m not going to go again through each section, I’ll give only one example to highlight what I mean; the text in lines 408-413 were we find one of those very messy and long sentences in which the authors try to include a lot of different information: “All *S. sclerotiorum* strains were cultured alone or together with an (in)compatibility strain on a cellophane membrane overlaying PDA with or without proline, or inoculated alone or co-inoculated on oilseed rape plants or *Arabidopsis*, and the mycelium was respectively collected and then total RNA was purified using a commercial RNA extraction kit (Takara, Code No.: 9108) for detection the expression of VIC-related genes with two methods, including the quantitative RT-PCR (RT413 qPCR) and RNA-seq.”

Response: Thank you very much, and we tried our best to use a simple sentence for our statement, please see lines 411-414.

In addition to being messy, the sentence includes odd wording like “for detection the expression of VIC...” Later all they say again “to detect the expression profile” (line 145); expression profiles are not detected, they can say for example to analyze or to study.

All these problems have to be fixed before the manuscript is in publishable form.

Response: Thank you very much for your efforts on our manuscript, and we tried our best to fix all the problems in the new version, please see lines 167-168; 242-244;429-431.

My final comment is regarding Figure 4D, I'm still concerned with the quality of these images and have my doubts they illustrate hyphal fusion as the authors claim. They authors have not changes the images so I guess they couldn't obtain others during the experiments that better illustrate their point. All authors have done is to add arrows pointing at the hyphae of the two strains and they added this comment: "the orientation of the white box indicates that the strain originally labeled with one fluorescent group showed a signal of another fluorescent group, suggesting the fusion of hyphae carrying different fluorescent labels". This sentence doesn't make sense to me. And all I see is what appears to be partial overlap of hyphae from different strains.

Response: we observed lots of pictures about the hyphal fusion process between incompatible or compatible strains under fluorescence microscope. It is indeed more difficult to observe the ideal hyhal fusion phenomenon in *Sclerotinia sclerotiorum* than other fungi, such as *Neurospora crassa*. We try our best to improve the quality of figures about hyphal fusion. Please see new figure 4D.

Reviewer #2 (Remarks to the Author):

My previous comments have been addressed.

Response: Thank you very much for your positive comment.